# *Caenorhabditis elegans* processes sensory information to choose between freeloading and self-defense strategies

Jodie A Schiffer[1], Francesco A Servello[1], William R Heath[1], Francis Raj Gandhi Amrit[2], Stephanie V Stumbur[1], Matthias Eder[3], Olivier MF Martin[3], Sean B Johnsen[1], Julian A Stanley[1], Hannah Tam[1], Sarah J Brennan[1], Natalie G McGowan[1], Abigail L Vogelaar[1], Yuyan Xu[1], William T Serkin[1], Arjumand Ghazi[2,4], Nicholas Stroustrup[3,5], Javier Apfeld[1]*

[1]Biology Department, Northeastern University, Boston, United States; [2]Department of Pediatrics, University of Pittsburgh School of Medicine, Pittsburgh, United States; [3]Centre for Genomic Regulation (CRG), The Barcelona Institute of Science and Technology, Barcelona, Spain; [4]Departments of Developmental Biology and Cell Biology and Physiology, University of Pittsburgh School of Medicine, Pittsburgh, United States; [5]Universitat Pompeu Fabra (UPF), Barcelona, Spain

**Abstract** Hydrogen peroxide is the preeminent chemical weapon that organisms use for combat. Individual cells rely on conserved defenses to prevent and repair peroxide-induced damage, but whether similar defenses might be coordinated across cells in animals remains poorly understood. Here, we identify a neuronal circuit in the nematode *Caenorhabditis elegans* that processes information perceived by two sensory neurons to control the induction of hydrogen peroxide defenses in the organism. We found that catalases produced by *Escherichia coli*, the nematode's food source, can deplete hydrogen peroxide from the local environment and thereby protect the nematodes. In the presence of *E. coli*, the nematode's neurons signal via TGFβ-insulin/ IGF1 relay to target tissues to repress expression of catalases and other hydrogen peroxide defenses. This adaptive strategy is the first example of a multicellular organism modulating its defenses when it expects to freeload from the protection provided by molecularly orthologous defenses from another species.

*For correspondence:
j.apfeld@northeastern.edu

Competing interests: The authors declare that no competing interests exist.

## Introduction

Bacteria, fungi, plants, and animal cells have long been known to excrete hydrogen peroxide to attack their prey and pathogens (*Avery and Morgan, 1924*). Hydrogen peroxide is also a byproduct of aerobic respiration (*Chance et al., 1979*). Cells rely on highly conserved defense mechanisms to degrade hydrogen peroxide and avoid the damage that hydrogen peroxide inflicts on their proteins, nucleic acids, and lipids (*Mishra and Imlay, 2012*). The extent to which these protective defenses are coordinated across cells in animals is poorly understood. In the present study, we used *C. elegans* as a model system to explore whether hydrogen peroxide protective defenses are coordinated across cells.

*C. elegans* is not spared from the threat of hydrogen peroxide. In its natural habitat of rotting fruits and vegetation, *C. elegans* encounters a wide variety of bacterial taxa (*Samuel et al., 2016*), and this community includes bacteria in many genera known to degrade or produce hydrogen peroxide (*Passardi et al., 2007*). Hydrogen peroxide produced by a bacterium from the *C. elegans* microbiome, *Rhizobium huautlense*, causes DNA damage to the nematodes (*Kniazeva and Ruvkun, 2019*), and many bacteria—including *S. pyogenes*, *S. pneumoniae*, *S. oralis*, and *E. faecium*—kill *C.*

**eLife digest** Cells of all kinds often wage chemical warfare against each other. Hydrogen peroxide is often the weapon of choice on the microscopic battlefield, where it is used to incapacitate opponents or to defend against attackers. For example, some plants produce hydrogen peroxide in response to infection to fight off disease-causing microbes.

Individual cells have also evolved defenses to prevent or repair 'injuries' caused by hydrogen peroxide. These are similar across many different species. They include enzymes called catalases, which break down hydrogen peroxide, and others to repair damage. However, scientists still do not fully understand how animals and other multicellular organisms might coordinate these defenses across their cells.

*Caenorhabditis elegans* is a microscopic species of worm that lives in rotting fruits. It often encounters the threat of cellular warfare: many types of bacteria in its environment generate hydrogen peroxide, and some can make enough to kill the worms outright. Like other organisms, *C. elegans* also produces catalases to defend itself against hydrogen peroxide attacks. However, it must activate its defenses at the right time; if it did so when they were not needed, this would result in a detrimental energy 'cost' to the worm.

Although *C. elegans* is a small organism containing only a defined number of cells, exactly why and how it switches its chemical defenses on or off remains unknown. Schiffer et al. therefore set out to determine how *C. elegans* controls these defenses, focusing on the role of the brain in detecting and processing information from its environment.

Experiments looking at the brains of genetically manipulated worms revealed a circuit of sensory nerve cells whose job is to tell the rest of the worm's tissues that they no longer need to produce defense enzymes. Crucially, the circuit became active when the worms sensed *E. coli* bacteria nearby. Bacteria in the same family as *E. coli* are normally found in in the same habitat as *C. elegans* and these bacteria are also known to make enzymes of their own to eliminate hydrogen peroxide around them. These results indicate that *C. elegans* can effectively decide, based on the activity of its circuit, when to use its own defenses and when to 'freeload' off those of neighboring bacteria.

This work is an important step towards understanding how sensory circuits in the brain can control hydrogen peroxide defenses in multicellular organisms. In the future, it could help researchers work out how more complex animals, like humans, coordinate their cellular defenses, and therefore potentially yield new strategies for improving health and longevity.

*elegans* by producing millimolar concentrations of hydrogen peroxide (*Bolm et al., 2004*; *Jansen et al., 2002*; *Moy et al., 2004*). *C. elegans* may also encounter hydrogen peroxide derived from fruits, leaves, and stems, because plants produce hydrogen peroxide to attack their pathogens (*Arakawa et al., 2014*; *Daudi et al., 2012*; *Mehdy, 1994*).

Coordinating hydrogen peroxide cellular defenses could be beneficial because it might enable *C. elegans* to avoid the energetic cost of unneeded protection. In addition, tight coordination of hydrogen peroxide defenses might be necessary because inducing a protective response at an inappropriate time might cause undesirable side effects. Hydrogen peroxide is an important intracellular signaling molecule, and depletion of hydrogen peroxide by scavenging enzymes may interfere with signal transduction and affect cell behavior and differentiation (*Veal et al., 2007*). Nematodes over-expressing all three catalase genes exhibit a high level of mortality due to internal hatching of larvae, and this phenotype can be suppressed by joint overexpression of the superoxide dismutase SOD-1 (*Doonan et al., 2008*), an enzyme that produces hydrogen peroxide. While catalases can degrade large quantities of hydrogen peroxide, at low hydrogen peroxide concentrations these enzymes accumulate in the ferryl-radical intermediate of their catalytic cycle, which is a dangerous oxidizing agent (*Imlay, 2013*).

We set out to investigate whether sensory neurons coordinate hydrogen peroxide protective defenses across cells because sensory circuits in the brain collect and integrate information from the environment, enabling animals to respond to environmental change. Specific sensory neurons enable nematodes to smell, taste, touch, and sense temperature and oxygen levels (*Bargmann and Horvitz, 1991a*; *Chalfie et al., 1985*; *Gray et al., 2004*; *Mori and Ohshima, 1995*; *White et al., 1986*).

This information is integrated rapidly by interneurons to direct the nematode's movement towards favorable environmental cues and away from harmful ones (*Kaplan et al., 2018*). Nematodes also use sensory information to modify their development, metabolism, lifespan, and heat defenses (*Apfeld and Kenyon, 1999*; *Bargmann and Horvitz, 1991b*; *Mak et al., 2006*; *Prahlad et al., 2008*). Understanding how sensory circuits in the brain regulate hydrogen peroxide defenses in *C. elegans* may provide a template for understanding how complex animals coordinate their cellular defenses in response to the perceived threat of hydrogen peroxide attack.

Using a systematic neuron-specific genetic-ablation approach, we identified ten classes of sensory neurons that regulate sensitivity to harmful peroxides in *C. elegans*. We found that the two ASI sensory neurons of the amphid, the major sensory organ of the nematode, initiate a multistep hormonal relay that decreases the nematode's hydrogen peroxide defenses: a DAF-7/TGFβ signal from ASI is received by multiple sets of interneurons, which independently process this information and then relay it to target tissues via insulin/IGF1 signals. Interestingly, this neuronal circuit lowers the action of endogenous catalases and other hydrogen peroxide defenses within the worm in response to perception and ingestion of *E. coli*, the nematode's primary food source in laboratory experiments. We show that *E. coli* express orthologous defenses that degrade hydrogen peroxide in the environment and that *C. elegans* does not need to induce catalases and other hydrogen peroxide defenses when *E. coli* is abundant. Thus, this neuronal circuit enables the nematodes to lower their own defenses upon sensing bacteria that can provide protection. In the microbial battlefield, nematodes use a sensory-neuronal circuit to determine whether to defend themselves from hydrogen peroxide attack or to freeload off protective defenses from another species.

## Results

### Sensory neurons regulate peroxide resistance in *C. elegans*

*C. elegans* is sensitive to the lethal effects of peroxides. Under standard laboratory conditions, wild-type nematodes have an average lifespan of approximately 15 days (*Kenyon et al., 1993*). In contrast, when grown in the presence of a peroxide (6 mM tert-butyl hydroperoxide, tBuOOH), the average lifespan of these nematodes is reduced to less than 1 day (*Figure 1A*; *An et al., 2005*). Previously, we determined the peroxide resistance of nematodes by measuring their lifespan with high temporal resolution in the presence of 6 mM tBuOOH (*Stroustrup et al., 2013*).

To investigate whether sensory neurons might regulate the nematode's peroxide defenses, we measured peroxide resistance in mutant animals with global defects in sensory perception. We first examined *osm-5* cilium structure mutants, which lack neuronal sensory perception due to defects in the sensory endings (cilia) of most sensory neurons (*Perkins et al., 1986*). These mutants exhibited a 45% increase in peroxide resistance relative to wild-type controls (*Figure 1A* and *Supplementary file 1*). Next, we examined *tax-2* and *tax-4* cyclic GMP-gated channel mutants, which are defective in the transduction of several sensory processes including smell, taste, oxygen, and temperature sensation (*Coburn and Bargmann, 1996*; *Komatsu et al., 1996*). These two mutants also exhibited large increases in peroxide resistance compared to wild-type controls (*Figure 1A*, *Figure 1—figure supplement 1*, and *Supplementary file 1*). Together, these observations indicate that neuronal sensory perception plays a role in regulating peroxide resistance in nematodes.

In *C. elegans* hermaphrodites, 60 ciliated and 12 non-ciliated neurons perform most sensory functions (*White et al., 1986*). To identify which of these sensory neurons influence the nematode's peroxide resistance, we systematically measured peroxide resistance in a collection of strains in which specific sensory neurons have been genetically ablated via neuron-specific expression of caspases (*Chelur and Chalfie, 2007*) or, in one case, via mutation of a neuron-specific fate determinant (*Chang et al., 2003*; *Uchida et al., 2003*). Overall, our neuron-ablation collection covered 44 ciliated and 10 non-ciliated neurons, including each of the 12 pairs of ciliated neurons that make up the two amphids (the major sensory organs), 8 of the 13 classes of non-amphid ciliated neurons, and 6 of the 7 classes of non-ciliated sensory neurons (*Supplementary file 10*). Individual ablation of ASI, ASG, ASK, AFD, AWC, IL2 and joint ablation of ADE, PDE, and CEP increased the nematode's peroxide resistance by up to 61% (*Figure 1A–B*, *Figure 1—figure supplement 1B–C*, and *Supplementary file 1*), whereas individual ablation of ASJ and AWA, and joint ablation of URX,

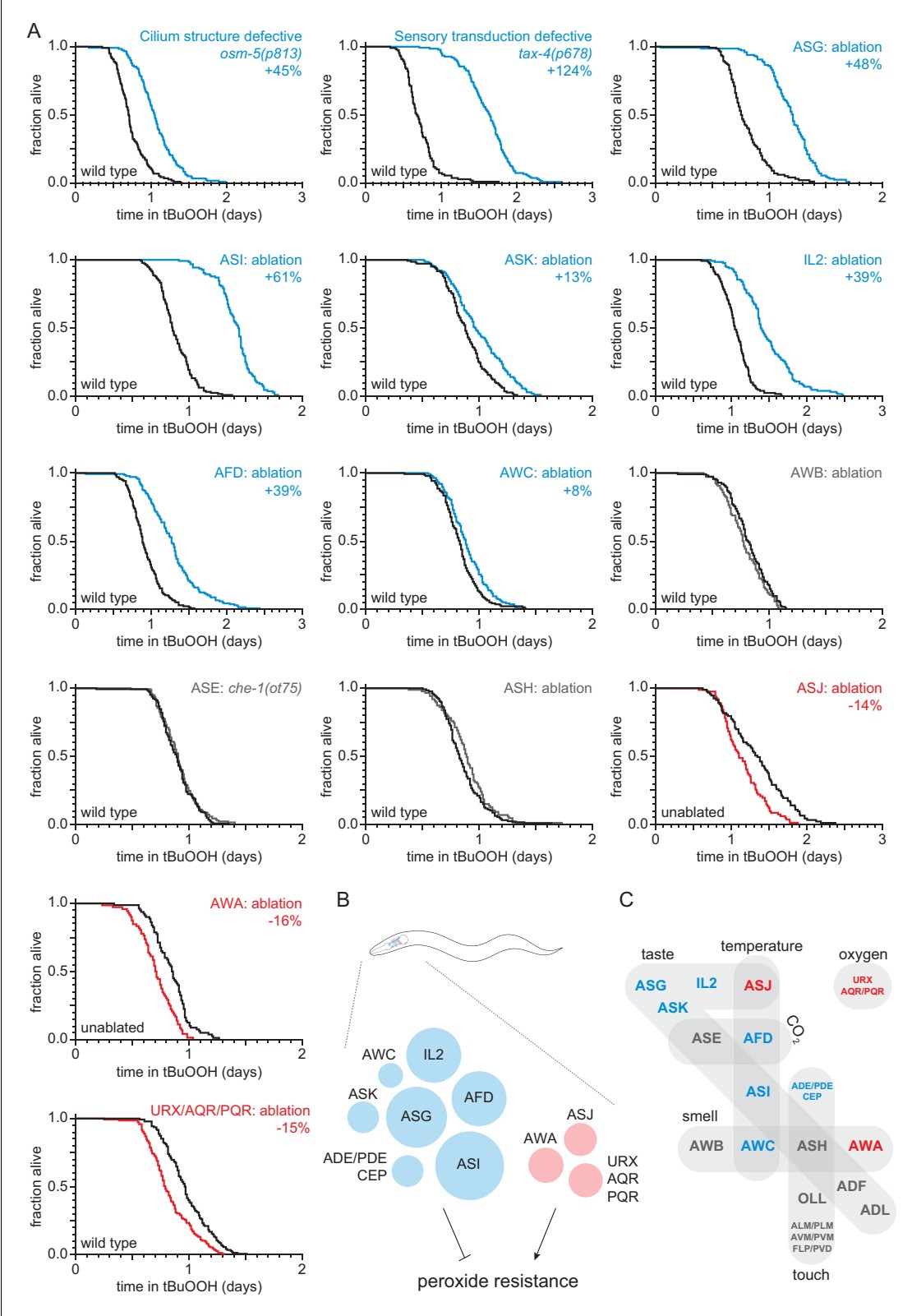

**Figure 1.** Sensory neurons regulate peroxide resistance in *C. elegans*. (**A**) Peroxide resistance of nematodes with defects in sensory cilia and sensory transduction, or with genetic ablation of specific sensory neurons. The fraction of nematodes remaining alive in the presence of 6 mM tert-butyl hydroperoxide (tBuOOH) is plotted against time. Interventions that increased, decreased, or did not affect survival are denoted in blue, red, and gray, respectively, and their effects on mean peroxide resistance are noted. (**B**) Specific sensory neurons normally reduce (blue) or increase (red) peroxide

*Figure 1 continued on next page*

*Figure 1 continued*

resistance. Circle area denotes the effect of ablation of the respective neurons on mean peroxide resistance. (**C**) Sensory neurons are grouped by the stimuli they sense. Neurons that normally reduce (seven classes) or increase (three classes) peroxide resistance are shown in blue and red, respectively. See also *Figure 1—figure supplement 1*. Statistical analyses are in *Supplementary file 1*.

The online version of this article includes the following figure supplement(s) for figure 1:

**Figure supplement 1.** Sensory neurons regulate peroxide resistance in *C. elegans*.

AQR, and PQR reduced peroxide resistance by up to 16% (*Figure 1A–B*). The remainder of the neurons tested—ADF, ADL, ASE, ASH, AWB, OLL, and joint ablation of ALM, PLM, AVM, PVM, FLP, and PVD—did not affect peroxide resistance (*Figure 1A and C*, *Figure 1—figure supplement 1D–G*, and *Supplementary file 1*). Altogether, we found that ten classes of sensory neurons can positively or negatively modulate peroxide resistance (*Figure 1B*). These neurons are known to respond to diverse stimuli, including smell, taste, touch, temperature, and oxygen levels (*Figure 1C*), suggesting that nematodes might adjust their peroxide resistance in response to multiple types of sensory information.

## ASI sensory neurons regulate peroxide resistance via DAF-7/TGFβ signaling

Among all neuronal ablations tested, ablation of ASI, a pair of neurons that sense taste and temperature, caused the largest increase in peroxide resistance (*Figure 1A*). Thus, we focused on the role of the ASI neuronal pair. ASI neurons secrete many peptide hormones, including DAF-7 (*Meisel et al., 2014*; *Ren et al., 1996*), a transforming growth factor β (TGFβ) hormone that regulates feeding, development, metabolism, and lifespan (*Dalfó et al., 2012*; *Greer et al., 2008*; *Ren et al., 1996*; *Shaw et al., 2007*). To determine whether DAF-7/TGFβ signaling also regulates peroxide resistance, we examined the effects of mutations in *daf-7*. We found that *daf-7(ok3125)* null and *daf-7(e1372)* loss-of-function mutations increased peroxide resistance two-fold relative to wild-type controls (*Figure 2A and B*, and *Figure 2—figure supplement 1A–D*, and *Supplementary file 2*). Reintroducing the *daf-7(+)* gene into *daf-7(ok3125)* mutants restored peroxide resistance to wild-type levels

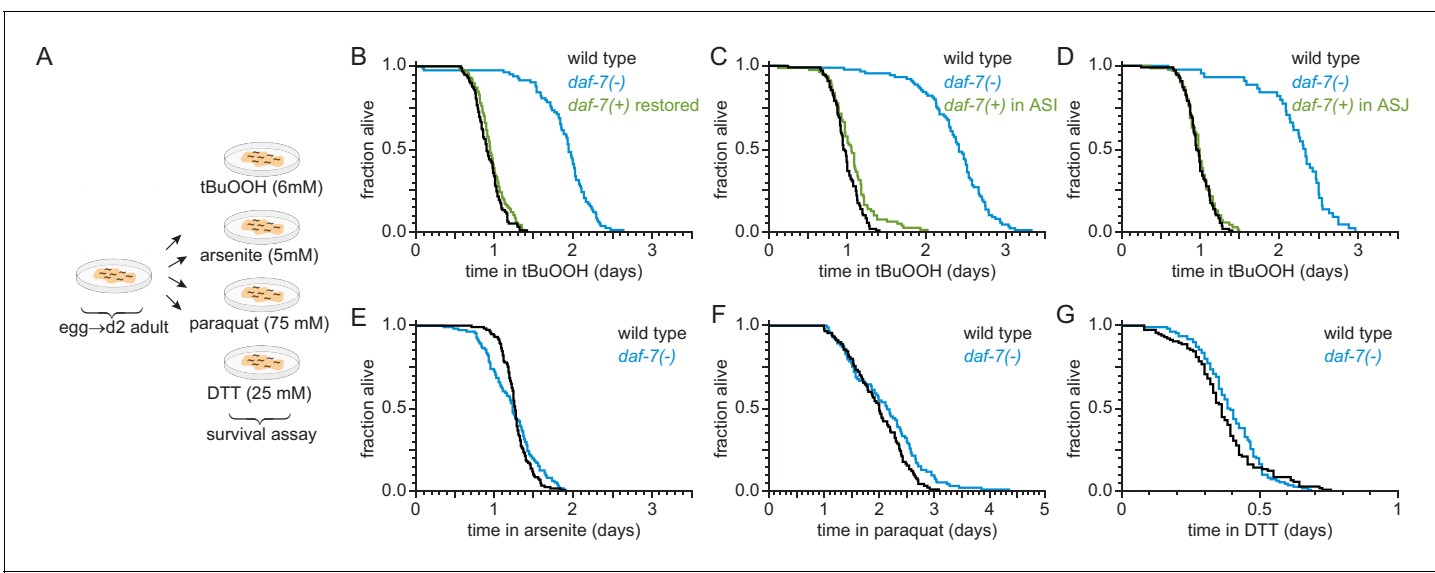

**Figure 2.** ASI sensory neurons secrete DAF-7/TGFβ to specifically lower the nematode's peroxide resistance. (**A**) Diagram summarizing experimental strategy. (**B–D**) Peroxide resistance of wild type, *daf-7(ok3125)*, and *daf-7(ok3125)* with *daf-7(+)* reintroduced with (**B**) its endogenous promoter, (**C**) the ASI-specific *str-3* promoter, or (**D**) the ASJ-specific *trx-1* promoter. (**E–G**) Resistance to 5 mM arsenite, 75 mM paraquat, and 25 mM dithiothreitol (DTT) of wild type and *daf-7(ok3125)*. See also *Figure 2—figure supplement 1*. Statistical analyses are in *Supplementary file 2*.

The online version of this article includes the following figure supplement(s) for figure 2:

**Figure supplement 1.** DAF-7 regulates peroxide resistance robustly.

(*Figure 2B* and *Supplementary file 2*). Moreover, expression of *daf-7(+)* only in the ASI neurons was sufficient to reduce the peroxide resistance of *daf-7(ok3125)* mutants to wild-type levels (*Figure 2C* and *Supplementary file 2*). *daf-7* is also expressed at a low level in ASJ, another pair of chemosensory neurons (*Meisel et al., 2014*), and expression of *daf-7(+)* only in ASJ rescued the increased peroxide resistance of *daf-7(ok3125)* mutants (*Figure 2D* and *Supplementary file 2*). Thus, expression of *daf-7* in ASI or ASJ was sufficient to confer normal peroxide resistance. Because ablation of ASI increased peroxide resistance but ablation of ASJ did not (*Figure 1A*), we reason that ASI neurons are the source of DAF-7 that regulates the nematode's peroxide resistance.

We next asked whether DAF-7/TGFβ from ASI might regulate resistance to additional toxic chemicals from the environment that are not peroxides or directly generate peroxides. We tested sensitivity of *daf-7* mutants to arsenite (a toxic metalloid), paraquat (a redox-cycling herbicide), and dithiothreitol/DTT (a reducing agent). We adjusted the concentrations of these compounds to reduce the survival of wild-type nematodes about as much as in the tBuOOH survival assays. Compared with wild-type animals, *daf-7(ok3125)* mutants had similar survival in 5 mM arsenite, 25 mM dithiothreitol, and 75 mM paraquat (*Figure 2A and E–G*, and *Supplementary file 2*). Therefore, the DAF-7/TGFβ signal from ASI is a specific regulator of peroxide resistance in the worm.

## The DAF-1/TGFβ receptor functions redundantly in interneurons to regulate peroxide resistance in response to DAF-7/TGFβ from ASI

DAF-7/TGFβ signals via the Type 1 TGFβ receptor DAF-1 (*Georgi et al., 1990*) to regulate multiple downstream processes (*Dalfó et al., 2012*; *Greer et al., 2008*; *Ren et al., 1996*; *Shaw et al., 2007*). Signaling through the DAF-1 receptor inactivates the transcriptional activity of a complex between the receptor-associated coSMAD, DAF-3, and the Sno/Ski factor, DAF-5 (*da Graca, 2004*; *Patterson et al., 1997*; *Tewari et al., 2004*). We found that a similar signal-transduction pathway regulates peroxide resistance. *daf-1(m40)* loss-of-function mutants showed a two-fold increase in peroxide resistance (*Figure 3A* and *Supplementary file 3*), and the increase in peroxide resistance of *daf-7* and *daf-1* mutants was almost completely abrogated by null or loss-of-function mutations in either *daf-3* or *daf-5* (*Figure 3A*, *Figure 3—figure supplement 1A–C*, and *Supplementary file 3*). The *daf-3(mgDf90)* null mutation also suppressed the increase in peroxide resistance of ASI-ablated worms (*Figure 3B* and *Supplementary file 3*). Therefore, the ASI neurons normally function to lower peroxide resistance in the worm using a canonical TGFβ signaling pathway.

To determine which cells receive the DAF-7/TGFβ signal from the ASI neurons to regulate peroxide resistance, we restored *daf-1(+)* gene expression in specific subsets of neurons using cell-type specific promoters in *daf-1(m40)* mutants. The cells composing each of these subsets of neurons, as well as the overlap between these subsets are diagramed in *Figure 3C*. DAF-1/TGFβ receptor is expressed broadly in the nervous system and in the distal-tip cells of the gonad (*Gunther et al., 2000*). Pan-neuronal expression of *daf-1(+)* with the *egl-3* promoter lowered peroxide resistance in *daf-1* mutants to the same extent as did expressing *daf-1(+)* with the endogenous *daf-1* promoter (*Figure 3D–E*, and *Supplementary file 3*). Reconstituting *daf-1(+)* expression in all ciliated neurons (except BAG and FLP) using the *osm-6* promoter had a minimal effect on peroxide resistance (*Figure 3F* and *Supplementary file 3*), indicating that *daf-1* function in ciliated neurons is not sufficient to lower peroxide resistance. In contrast, expression of *daf-1(+)* in multiple sets of non-ciliated interneurons and pharyngeal neurons using the *flp-1*, *tdc-1*, *glr-1*, or *glr-8* promoters lowered peroxide resistance to a similar extent as pan-neuronal expression of *daf-1(+)* in *daf-1* mutants (*Figure 3G–I*, *Figure 3—figure supplement 1D*, and *Supplementary file 3*), while directed *daf-1(+)* expression in nine pharyngeal neurons using the *glr-7* promoter did not affect peroxide resistance (*Figure 3—figure supplement 1E* and *Supplementary file 3*). The *flp-1* promoter is active only in the two AVK interneurons (*Greer et al., 2008*). In addition, the *flp-1*, *tdc-1*, *glr-1*, and *glr-8* promoters drive expression in non-overlapping cells, except for the expression overlap in the two RIM interneurons by the *tdc-1* and *glr-1* promoters (*Greer et al., 2008*; *Figure 3C*). We refer to the sets of neurons where *flp-1*, *tdc-1*, *glr-1*, and *glr-8* are expressed as 'DAF-1-sufficiency sets', because expression of *daf-1(+)* in any one of these sets of neurons is sufficient to lower the peroxide resistance of *daf-1* mutant nematodes. We conclude that DAF-1 functions redundantly in AVK interneurons and at least two other separate sets of neurons to lower the nematode's peroxide resistance.

Where does the DAF-3/coSMAD transcription factor function to promote peroxide resistance when the DAF-1/TGFβ-receptor is inactive? We expected that DAF-3 would function in the same

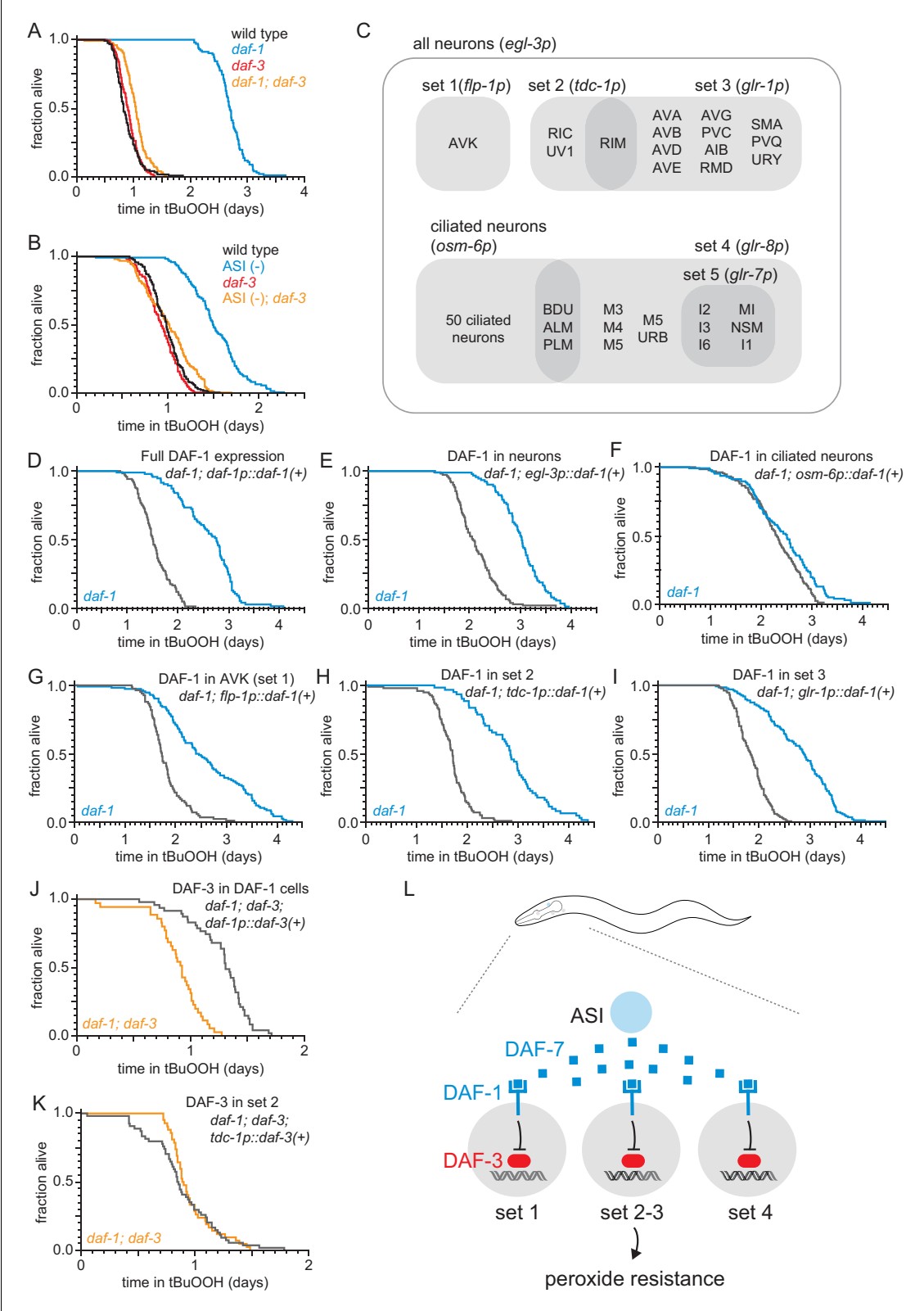

**Figure 3.** The DAF-1/TGFβ receptor functions redundantly in interneurons to regulate peroxide resistance in response to DAF-7/TGFβ from ASI. (A–B) *daf-3(mgDf90)* almost completely abrogated the increased peroxide resistance of (A) *daf-1(m40)* and of (B) genetic ablation of ASI. (C) Diagram of the subsets of neurons where *daf-1(+)* or *daf-3(+)* were expressed in transgenic rescue experiments shown in panels (D–K) and in *Figure 3—figure supplement 1*. The promoter elements used to drive gene expression in those subsets of neurons are shown in parentheses. The *tdc-1* promoter also

*Figure 3 continued on next page*

Figure 3 continued

drives gene expression in the sheath cells of the somatic gonad. (D–I) Peroxide resistance of transgenic nematodes expressing *daf-1(+)* in specific subsets of cells and *daf-1(m40)* controls. (J–K) Peroxide resistance of transgenic nematodes expressing *daf-3(+)* in specific subsets of cells and *daf-1 (m40); daf-3(mgDf90)* controls. (L) ASI signals to three sets of interneurons to lower the nematode's peroxide resistance. To increase peroxide resistance, all of these sets of neurons must independently activate the DAF-3/DAF-5 transcriptional complex. See also *Figure 3—figure supplement 1*. Statistical analyses are in *Supplementary file 3*.

The online version of this article includes the following figure supplement(s) for figure 3:

**Figure supplement 1.** The DAF-1/TGFβ receptor functions redundantly in interneurons to regulate peroxide resistance in response to DAF-7/TGFβ from ASI.

cells as DAF-1 to regulate peroxide resistance, because both of these canonical TGFβ signal-transduction pathway components function in *tdc-1* expressing interneurons to regulate feeding, fat storage, egg laying, and dauer-larva formation (*Greer et al., 2008*). In addition, because during signal transduction DAF-1 inhibits DAF-3, we expected that when DAF-1 is active only in one set of neurons then DAF-3 should be active only in neurons outside that set (including the neurons of other non-overlapping DAF-1-sufficiency sets). This implied that to increase peroxide resistance DAF-3 should be active in all DAF-1-sufficiency sets of neurons. To test that prediction, we examined the effect on peroxide resistance of restoring *daf-3(+)* expression in just one of the DAF-1-sufficiency sets of neurons in *daf-1; daf-3* double mutants. Confirming our prediction, we found that restoring *daf-3(+)* expression with the *tdc-1* promoter was not sufficient to increase peroxide resistance in *daf-1; daf-3* double mutants (*Figure 3K*, *Figure 3—figure supplement 1F*, and *Supplementary file 3*). In contrast, the peroxide resistance of *daf-1; daf-3* double mutants increased upon restoring *daf-3(+)* expression in all four DAF-1-sufficiency sets of neurons with a *daf-1* promoter (*Figure 3J* and *Supplementary file 3*). We propose that the combination of the redundant action of DAF-1 in multiple sets of neurons and the repression of DAF-3 by DAF-1 in each of those neurons ensures that the nematode's peroxide resistance stays low until all DAF-1-sufficiency sets of neurons de-repress DAF-3/coSMAD (*Figure 3L*). Alternatively, to promote peroxide resistance in animals with reduced DAF-1 activity, DAF-3 function may be necessary only in cells that do not express the *tdc-1* promoter. In such a scenario, other signaling molecules would transduce DAF-1 activity in *tdc-1*-expressing neurons to regulate peroxide resistance.

## ASI regulates the nematode's peroxide resistance via a TGFβ-Insulin/IGF1 hormone relay

Previous studies have shown that different mechanisms are required downstream of the DAF-3/coSMAD transcription factor to mediate the effects of DAF-7/TGFβ signaling on dauer-larva formation, fat storage, germline size, lifespan, and feeding (*Dalfó et al., 2012*; *Greer et al., 2008*; *Shaw et al., 2007*). In this section, and later in this manuscript, we used a genetic approach to determine whether DAF-7/TGFβ signaling acts via one or more of these mechanisms to regulate the nematode's peroxide resistance (*Figure 4A*).

DAF-7 regulates dauer-larva formation via the nuclear hormone receptor DAF-12, which is the main switch driving the choice of reproductive growth or dauer arrest (*Antebi et al., 2000*). Loss of *daf-12* suppresses the constitutive dauer-formation phenotype of *daf-7* loss-of-function mutants during development (*Thomas et al., 1993*), but the *daf-12(rh61rh411)* null mutation did not suppress the increased peroxide resistance of *daf-7(ok3125)* null adults (*Figure 4B* and *Supplementary file 4*). In fact, even though the *daf-12* null mutation lowered the peroxide resistance in otherwise wild-type animals, it further increased peroxide resistance in *daf-7* mutants. We conclude that *daf-12(+)* limits the peroxide resistance of *daf-7* mutants, and that DAF-7 lowers peroxide resistance and inhibits formation of peroxide-resistant dauer larvae via separate mechanisms.

The metabotropic glutamate receptors *mgl-1* and *mgl-3* are necessary for the increase in fat storage upon DAF-7-pathway inhibition (*Greer et al., 2008*). However, null mutations in either or both of these *mgl* genes did not affect peroxide resistance in *daf-1* mutants (*Figure 4C*, *Figure 4—figure supplement 1*, and *Supplementary file 4*). Thus, peroxide resistance and fat storage are also regulated via separate pathways downstream of DAF-1.

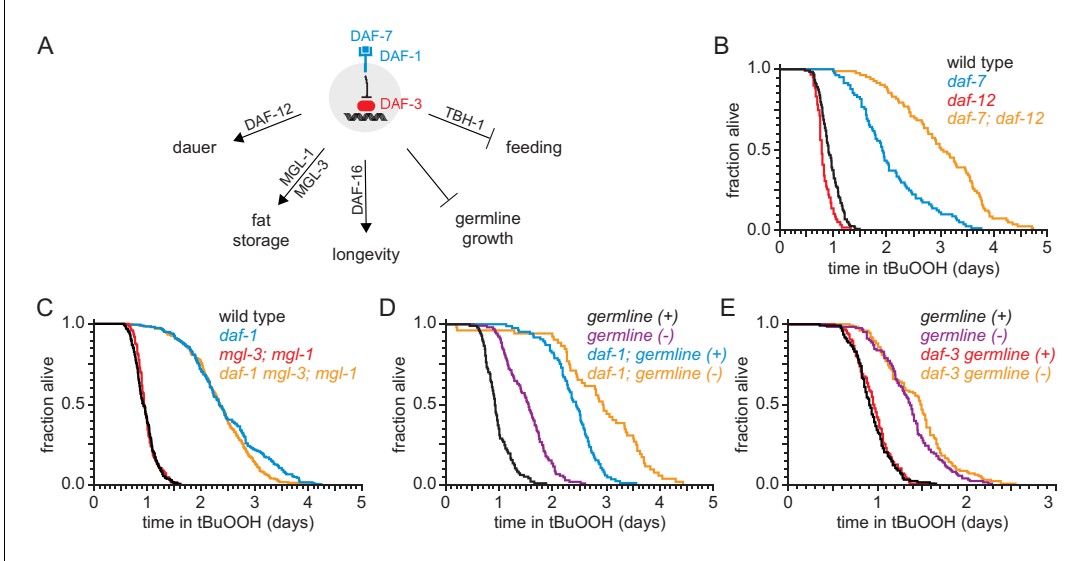

**Figure 4.** DAF-1/TGFβ-receptor signaling regulates peroxide resistance separately from its role in dauer formation, fat storage, and germline growth. (A) Different mechanisms operate downstream of DAF-3 to mediate the effects of DAF-7/TGFβ signaling on dauer-larva formation, fat storage, germline size, lifespan, and feeding. (B) *daf-12(rh61rh411)* did not suppress the increased peroxide resistance of *daf-7(e1372)*. (C) *mgl-1(tm1811)* and *mgl-3(tm1766)* did not jointly suppress the increased peroxide resistance of *daf-1(m40)*. (D) Genetic ablation of the germline and *daf-1(m40)* independently increased peroxide resistance. (E) *daf-3(mgDf90)* did not suppress the increased peroxide resistance of genetic ablation of the germline. See also *Figure 4—figure supplement 1*. Statistical analyses are in *Supplementary file 4*.

The online version of this article includes the following figure supplement(s) for figure 4:

**Figure supplement 1.** DAF-1/TGFβ-receptor signaling regulates peroxide resistance separately from its role in fat storage.

Germline size is reduced upon DAF-7-pathway inhibition (*Dalfó et al., 2012*). Mutations in the *mes-1* gene cause about 50% of animals to become sterile adults because they fail to form the primordial germ cells during embryogenesis, while the remaining animals develop into fertile adults (*Strome et al., 1995*). Germline-ablated *mes-1(ok2467)* mutants showed a 57% increase in peroxide resistance compared to their fertile *mes-1(ok2467)* mutant siblings (*Figure 4D* and *Supplementary file 4*), consistent with previous studies (*Steinbaugh et al., 2015*). However, *daf-1 (m40)* increased peroxide resistance in both germline-ablated and fertile *mes-1* mutants (*Figure 4D* and *Supplementary file 4*). In addition, *daf-3(mgDf90)* did not affect peroxide resistance in germline-ablated *mes-1* mutants (*Figure 4E* and *Supplementary file 4*). Therefore, DAF-1 and the germline regulate peroxide resistance via independent mechanisms.

DAF-7-pathway signaling lowers lifespan by promoting insulin/IGF1 receptor signaling (*Shaw et al., 2007*). Previous studies have shown that transcription of at least 11 of the 40 insulin/IGF1 genes in the genome is repressed by the DAF-3/coSMAD in response to lower levels of DAF-7 and DAF-1 signaling (*Liu et al., 2004*; *Narasimhan et al., 2011*; *Shaw et al., 2007*). We found that deletion of the DAF-3-repressed insulin/IGF1 genes *ins-1*, *ins-3*, *ins-4*, *ins-5*, *ins-6*, or *daf-28* caused increases in peroxide resistance ranging between 11% and 65% (*Figure 5A*, *Figure 5—figure supplement 1A–B*, and *Supplementary file 5*), suggesting DAF-7 lowers peroxide resistance by promoting signaling by the insulin/IGF1 receptor, DAF-2. The *daf-2(e1370)* strong loss-of-function mutation increased peroxide resistance about three-fold (*Figure 5B* and *Supplementary file 5*), consistent with previous findings (*Tullet et al., 2008*). Double mutants of *daf-1(m40)* and *daf-2(e1370)* had higher peroxide resistance than the respective single mutants (*Figure 5B* and *Supplementary file 5*). This additive effect suggested that the DAF-1 TGFβ receptor and the DAF-2 insulin/IGF1 receptor regulated peroxide resistance via mechanisms that do not fully overlap, but could also have been due to the receptors acting via fully overlapping mechanisms (because neither *daf-1(m40)* nor *daf-2(e1370)* eliminates gene function completely). We considered the possibility that a DAF-2-dependent mechanism might mediate some of the effects of DAF-1 on peroxide resistance. If repressing the expression of insulin/IGF1 ligands of DAF-2 mediated part of the increased

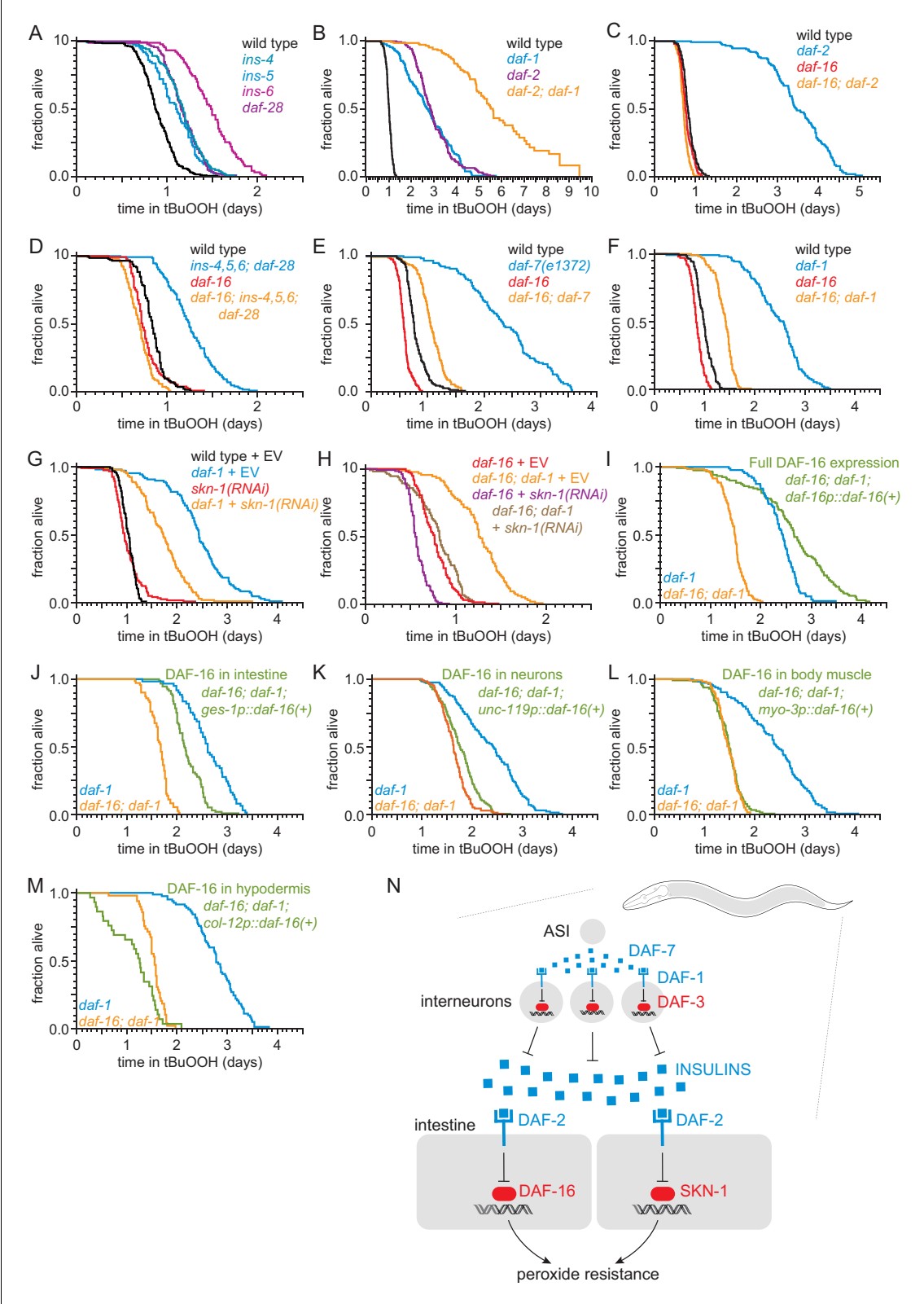

**Figure 5.** ASI regulates the nematode's peroxide resistance via a TGFβ-Insulin/IGF1 hormone relay. (**A**) Deletions in *ins-4*, *ins-5*, *ins-6*, and *daf-28* insulin-coding genes increased peroxide resistance. (**B**) *daf-2(e1370)* and *daf-1(m40)* independently increased peroxide resistance. (**C–D**) *daf-16(mu86)* abrogated the increased peroxide resistance of (**C**) *daf-2(e1370)* and (**D**) an *ins-4 ins-5 ins-6; daf-28* quadruple mutant. (**E–F**) *daf-16(mu86)* suppressed part of the increased peroxide resistance of (**E**) *daf-7(e1372)* and (**F**) *daf-1(m40)*. (**G**) *skn-1(RNAi)* suppressed part of the increased peroxide resistance of

*Figure 5 continued on next page*

*Figure 5 continued*

*daf-1(m40)*. Control RNAi consisted of feeding the nematodes the same bacteria but with the empty vector (EV) plasmid pL4440 instead of a plasmid targeting *skn-1*. (H) *skn-1(RNAi)* lowered the peroxide resistance of *daf-16(mu86); daf-1(m40)*. (I–M) Peroxide resistance of transgenic nematodes expressing *daf-16(+)* in specific subsets of cells, *daf-16(mu86); daf-1(m40)* controls, and *daf-1(m40)* reference. (N) ASI sensory neurons make nematodes more sensitive to hydrogen peroxide via a multistep hormonal relay. DAF-7/TGFβ from ASI is received by interneurons. These interneurons act redundantly to relay this signal to target tissues by promoting transcription of insulin genes. These insulins activate the DAF-2 insulin/IGF1 receptor, leading to inhibition of DAF-16-dependent peroxide protection services by the intestine and neurons. SKN-1 acts independently of DAF-16 to promote peroxide resistance in response to reduced DAF-1 signaling. SKN-1 likely acts in the intestine, because *skn-1(+)* promotes peroxide resistance in *daf-2* mutants and induces oxidative-stress defenses in this tissue (*An et al., 2005*; *Tullet et al., 2008*). See also *Figure 5—figure supplement 1*. Statistical analyses are in *Supplementary file 5*.

The online version of this article includes the following figure supplement(s) for figure 5:

**Figure supplement 1.** ASI regulates the nematode's peroxide resistance via a TGFβ-Insulin/IGF1 hormone relay.

peroxide resistance of DAF-7-pathway inhibition, then one would expect the FOXO transcription factor DAF-16 to be necessary for those effects. DAF-16 is necessary for the increase in lifespan and most other phenotypes of mutants with reduced signaling by the DAF-2 insulin/IGF1 receptor (*Kenyon et al., 1993*; *Lin et al., 1997*; *Ogg et al., 1997*). We found that DAF-16 was also necessary for the increase in peroxide resistance of *daf-2(e1370)* mutants (*Figure 5C* and *Supplementary file 5*) and for the increase in peroxide resistance of an *ins-4 ins-5 ins-6; daf-28* quadruple mutant (*Figure 5D* and *Supplementary file 5*). The *daf-16(mu86)* null mutation decreased the peroxide resistance of *daf-7(e1372)* and *daf-1(m40)* mutants by nearly 50%, but caused only a small peroxide resistance reduction in wild-type nematodes (*Figure 5E–F*, and *Supplementary file 5*). Therefore, regulation of peroxide resistance by the DAF-7/TGFβ signaling pathway is, in part, dependent on the DAF-16/FOXO transcription factor.

We examined whether other transcription factors might act with DAF-16 to increase peroxide resistance in *daf-1* mutants. Like DAF-16, the NRF orthologue SKN-1 and the TFEB orthologue HLH-30 are activated in response to reduced DAF-2 signaling (*Lin et al., 2018*; *Tullet et al., 2008*). The peroxide resistance of *daf-1(m40) hlh-30(tm1978)* double mutants was identical to that of *daf-1* single mutants (*Figure 5—figure supplement 1C* and *Supplementary file 5*). Knockdown of *skn-1* via RNA interference (RNAi) decreased the peroxide resistance of *daf-1(m40)* mutants by 30% but did not affect peroxide resistance in wild-type nematodes (*Figure 5G* and *Supplementary file 5*). RNAi of *skn-1* also decreased the peroxide resistance of *daf-16; daf-1* double mutants, suggesting that DAF-16 and SKN-1 functioned in a non-overlapping manner to promote peroxide resistance in *daf-1 (m40)* mutants (*Figure 5H* and *Supplementary file 5*). We propose that repression of insulin/IGF1 gene expression by DAF-3/coSMAD leads to a reduction in signaling by the DAF-2/insulin/IGF1 receptor, which subsequently increases the nematode's peroxide resistance via transcriptional activation by SKN-1/NRF and DAF-16/FOXO (*Figure 5N*).

## DAF-16/FOXO functions in intestine and neurons to increase the nematode's peroxide resistance in response to reduced DAF-7/TGFβ signaling

To identify which target tissues are important for increasing the nematode's peroxide resistance via DAF-16 in response to reduced DAF-1 signaling, we determined the extent to which restoring *daf-16(+)* expression in specific tissues using tissue-specific promoters increased the peroxide resistance of *daf-16; daf-1* double mutants. As expected, peroxide resistance was increased when we restored *daf-16(+)* expression with the endogenous *daf-16* promoter (*Figure 5I* and *Supplementary file 5*). Restoring *daf-16(+)* expression only in the intestine increased peroxide resistance, albeit to a lesser extent than did re-expressing *daf-16(+)* with the endogenous *daf-16* promoter (*Figure 5J* and *Supplementary file 5*). Restoring *daf-16(+)* in neurons slightly increased peroxide resistance (*Figure 5K*), while restoring *daf-16(+)* in body-wall muscles had no effect (*Figure 5L* and *Supplementary file 5*). Restoring *daf-16(+)* expression in the hypodermis decreased peroxide resistance slightly (*Figure 5M* and *Supplementary file 5*); however, it is difficult to interpret these results because these nematodes looked sickly (unlike *daf-1* and *daf-16* single and double mutants), consistent with reports that selectively expressing *daf-16(+)* in the hypodermis is toxic (*Libina et al., 2003*). Therefore, the DAF-16/FOXO transcription factor functions in the intestine and neurons to

increase the nematode's peroxide resistance when DAF-3/coSMAD is active due to reduced DAF-1 function (**Figure 5N**).

## Reduced DAF-7/TGFβ signaling upregulates expression of DAF-16/FOXO and SKN-1/NRF target genes

To investigate how reduced DAF-7/TGFβ signaling increases peroxide resistance, we used mRNA sequencing (mRNA-seq) to identify genes that were differentially regulated between *daf-7(ok3125)* mutants and wild-type animals. We extracted mRNA from day two adults and then performed differential expression analyses on the mRNA-seq data for the 9660 genes that had detectable expression. Relative to wild-type animals, *daf-7(ok3125)* null mutants decreased the expression of 3641 genes and increased the expression of 3229 genes (q value < 0.001) (**Figure 6A** and **Figure 6—figure supplement 1A**). These changes in gene expression were consistent with but more extensive than those observed in microarray-based studies with partial loss-of-function TGFβ signaling pathway mutants (**Shaw et al., 2007**; **Figure 6—figure supplement 1B** and **Supplementary file 6**). To identify which processes may be influenced by the transcriptomic changes of *daf-7* null mutants, we used Gene Ontology (GO) term enrichment analysis (**Angeles-Albores et al., 2016**) and clustered enriched GO terms based on semantic similarity (**Supek et al., 2011**). We focused the GO analysis

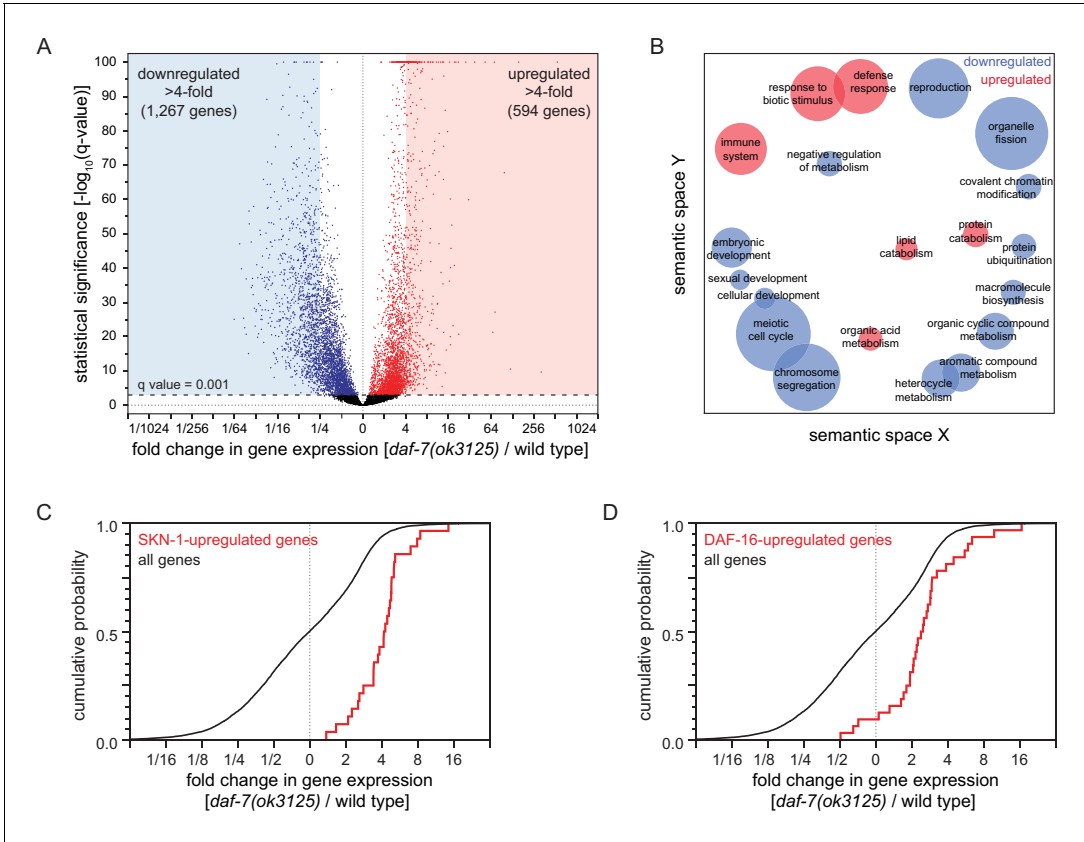

**Figure 6.** Reduced DAF-7/TGFβ signaling upregulates expression of DAF-16/FOXO and SKN-1/NRF target genes. (**A**) Volcano plot showing the level and statistical significance of changes in gene expression induced by the *daf-7(ok3125)* null mutation. Genes up- and down-regulated significantly (q value < 0.001) are shown in red and blue, respectively. (**B**) Gene Ontology (GO) term enrichment analysis of biological processes associated with the set of 594 upregulated genes (blue bubbles) and the set of 1267 downregulated genes (red bubbles) with a statistically significant and greater than four-fold change in expression in *daf-7(ok3125)* mutants relative to wild-type animals. Bubble size is proportional to the statistical significance [-log₁₀($P$ value)] of enrichment. (**C**) The *daf-7(ok3125)* mutation increased the expression of genes upregulated by *skn-1(+)* in wild type animals (**Oliveira et al., 2009**). (**D**) The *daf-7(ok3125)* mutation increased the expression of genes directly upregulated by DAF-16 (**Kumar et al., 2015**). See also **Figure 6—figure supplement 1**. Additional statistical analyses are in **Supplementary file 6**.

The online version of this article includes the following figure supplement(s) for figure 6:

**Figure supplement 1.** Reduced DAF-7/TGFβ signaling upregulates expression of DAF-16/FOXO and SKN-1/NRF target genes.

on genes upregulated or downregulated more than four-fold. The 1267 genes downregulated more than four-fold in *daf-7(ok3125)* mutants were associated with reproduction and with expression in the germline, while the 594 genes upregulated more than four-fold in *daf-7(ok3125)* mutants were associated with defense and immune responses and with expression in the intestine (*Figure 6B*).

Because the *skn-1* and *daf-16* genes were each partially required for the increased peroxide resistance of animals with reduced DAF-7/TGFβ signaling, we expected that the expression of SKN-1 and DAF-16 transcriptional targets would be influenced by *daf-7*. The *daf-7(ok3125)* mutation increased the expression of genes upregulated by *skn-1(+)* in wild type animals (*Oliveira et al., 2009*) and in *daf-2(-)* mutants (*Ewald et al., 2015*; *Figure 6C*, *Figure 6—figure supplement 1C*, and *Supplementary file 6*). In addition, the *daf-7(ok3125)* mutation increased the expression of genes directly upregulated by DAF-16 (*Kumar et al., 2015*; *Figure 6C* and *Supplementary file 6*) and, as observed in a previous study (*Shaw et al., 2007*), increased the expression of genes upregulated in a *daf-16*-dependent manner in *daf-2(-)* mutants (*Murphy et al., 2003*; *Figure 6—figure supplement 1D* and *Supplementary file 6*). Together, these findings suggest that DAF-7/TGFβ represses the induction of direct DAF-16 and SKN-1 target genes.

## Food ingestion regulates the nematode's peroxide resistance via DAF-3/coSMAD

Nematodes can be exposed directly to peroxides through food ingestion, and *daf-7* and *daf-1* mutants have been shown to exhibit mild feeding defects (*Greer et al., 2008*). Therefore, we considered the possibility that the increase in peroxide resistance of mutants with impaired DAF-7-pathway signaling was due to their reduced feeding. Previous studies have shown that the tyrosine decarboxylase TDC-1 and the tyramine β-hydroxylase TBH-1—biosynthetic enzymes for the neurotransmitters tyramine and octopamine, respectively—are each necessary for the feeding defect of *daf-1* mutants as *daf-1;tdc-1* and *daf-1;tbh-1* double mutants have normal feeding behaviors (*Greer et al., 2008*). Surprisingly, despite restoring normal feeding to *daf-1* mutants, *tbh-1* and *tdc-1* null mutations did not suppress the increased peroxide resistance of *daf-1* mutants (*Figure 7A*, *Figure 7—figure supplement 1A*, and *Supplementary file 7*). In fact, both mutations further increased peroxide resistance in a *daf-1* mutant background. Because mutations that restored normal feeding to *daf-1* mutants increased the peroxide resistance of *daf-1* mutants, these findings suggested that the reduced feeding exhibited by *daf-1* mutants in fact reduces the magnitude of their increased peroxide resistance.

To investigate whether feeding has a direct effect on peroxide resistance, we first determined whether a wild-type nematode's feeding history (before peroxide exposure) might affect its subsequent peroxide resistance. We transferred nematodes to plates with different concentrations of *E. coli* for 24 hr prior to the start of the peroxide resistance assay and found that the *E. coli* concentration before the assay had a dose-dependent effect on peroxide resistance (*Figure 7B* and *Supplementary file 7*). Animals grown on higher concentrations of *E. coli* had higher peroxide resistance. Strikingly, nematodes grown without *E. coli* for two days before the assay showed a six-fold decrease in peroxide resistance (*Figure 7C*, *Figure 7—figure supplement 1B*, and *Supplementary file 7*), even though they had access to plentiful *E. coli* during the assay.

Next, we tested whether reduced ingestion of *E. coli* was sufficient to mimic the effects of pre-exposure to reduced *E. coli* levels. Mutants in the pharyngeal-muscle specific nicotinic acetylcholine receptor subunit *eat-2* ingest bacteria more slowly due to reduced pharyngeal pumping (feeding) (*Avery, 1993*; *Raizen et al., 1995*). The *eat-2(ad1116)* loss-of-function mutation, which causes a strong feeding defect (*Figure 7E*), decreased peroxide resistance by 25% relative to wild-type animals (*Figure 7D* and *Supplementary file 7*). Therefore, impaired feeding leads to decreased peroxide resistance.

Finally, we asked whether feeding and DAF-7 signaling regulate peroxide resistance jointly, or independently. Unlike *eat-2* mutants, *daf-3* null single mutants did not decrease peroxide resistance compared with wild-type animals (*Figure 3A–B* and *Figure 6D*). However, the *eat-2(ad1116)* mutation caused a larger decrease in peroxide resistance in *daf-3* mutants than in wild-type nematodes (*Figure 7D* and *Supplementary file 7*), suggesting that *daf-3(+)* promotes peroxide resistance in *eat-2* mutants. This effect was not due to an enhancement of the feeding defect of *eat-2* mutants by the *daf-3* mutation, because *eat-2; daf-3* double mutants fed slightly more (not less) than *eat-2* single mutants (*Figure 7E*). We propose that DAF-3 is activated in response to reduced feeding, leading to

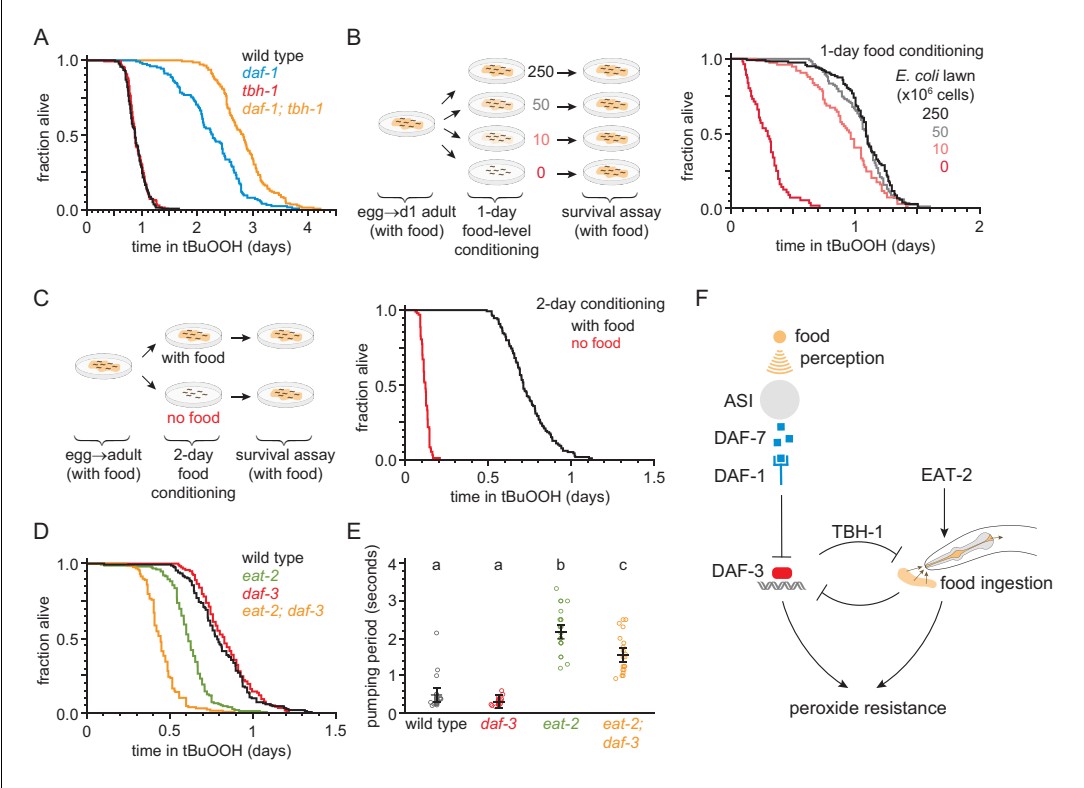

**Figure 7.** Food ingestion regulates the nematode's peroxide resistance via DAF-3/coSMAD. (A) *tbh-1(ok1196)* increased the peroxide resistance of *daf-1(m40)*. (B–C) The *E. coli* level before the assay affected *C. elegans* peroxide resistance in a dose-dependent manner. (D) *eat-2(ad1116)* caused a more severe reduction in peroxide resistance in *daf-3(mgDf90)* than in wild type. (E) *eat-2(ad1116)* caused a less severe reduction in feeding in *daf-3(mgDf90)* than in wild type. Lines mark the mean pumping period and its 95% confidence interval. Genotypes labeled with different letters exhibited significant differences in pumping period (p < 0.0001, Turkey HSD test) otherwise (p > 0.05). (F) DAF-3 and feeding increase peroxide resistance but attenuate each other's effects. Feeding inhibits DAF-3; this attenuates the reduction in peroxide resistance caused by reduced feeding. DAF-3 inhibits feeding via TBH-1; this attenuates the increase in peroxide resistance of *daf-1* mutants. Sensory perception of *E. coli* induces DAF-7 expression (*Chang et al., 2006*; *Gallagher et al., 2013*) in a concentration-dependent manner (*Entchev et al., 2015*; *Ren et al., 1996*) leading to DAF-3 repression by the DAF-7 receptor DAF-1. Therefore, both ingestion and perception of *E. coli* inhibit DAF-3. See also *Figure 7—figure supplement 1*. Additional statistical analyses are in *Supplementary file 7*.

The online version of this article includes the following figure supplement(s) for figure 7:

**Figure supplement 1.** DAF-3/coSMAD increases the nematode's peroxide resistance in response to reduced feeding.

an increase in peroxide resistance. DAF-3 acts as an adaptive mechanism that partially offsets the detrimental effect of reduced feeding on peroxide resistance.

Taken together, these findings imply that both feeding on bacteria and DAF-3/coSMAD signaling increase peroxide resistance, but that they attenuate each other's effects (*Figure 7F*). This cross-inhibition might enable nematodes to switch between DAF-3-dependent and DAF-3-independent mechanisms of peroxide resistance in response to changes in food ingestion and DAF-7 signal levels.

## DAF-7/TGFβ signals that hydrogen peroxide protection will be provided by catalases from *E. coli* and not by catalases from *C. elegans*

Why does DAF-7 from ASI function to decrease the nematode's peroxide resistance? ASI sensory neurons become active in response to perception of water-soluble signals from *E. coli* (*Gallagher et al., 2013*) and induce *daf-7* gene expression in a TAX-4-activity-dependent manner (*Chang et al., 2006*). As a result, the ASI neurons upregulate *daf-7* expression in response to *E. coli* (*Gallagher et al., 2013*) and lower *daf-7* gene expression in response to starvation and low *E. coli* concentrations (*Entchev et al., 2015*; *Ren et al., 1996*). Lowering DAF-7 levels when *E. coli* is scarce may enable nematodes to prepare for a future of reduced feeding by attenuating the expected

reduction in peroxide resistance caused by reduced feeding. But increasing DAF-7 levels when *E. coli* is abundant may render nematodes more vulnerable to peroxide. We reasoned that perhaps *C. elegans* decreases its own peroxide self-defenses via DAF-7 signaling from the ASI neurons when *E. coli* is abundant because *C. elegans* expects to be safe from peroxide attack in that setting.

To test that hypothesis, we first asked whether *E. coli* can protect nematodes from the lethal effects of peroxides. This required that we re-examine the conditions of the peroxide resistance assays, which were conducted using a lipid hydroperoxide (tert-butyl hydroperoxide, tBuOOH) widely used in *C. elegans* studies due to its stability (*An et al., 2005*). When we used hydrogen peroxide instead of tBuOOH, we could not kill *C. elegans* even with concentrations as high as 20 mM (*Figure 8A*) which is well above the biologically plausible range of up to 3 mM hydrogen peroxide used by other bacteria to kill *C. elegans* (*Bolm et al., 2004*; *Moy et al., 2004*). This suggested that hydrogen peroxide, but not tBuOOH, was efficiently degraded by *E. coli*. This bacterium uses several scavenging systems to degrade hydrogen peroxide (*Mishra and Imlay, 2012*). The two *E. coli* catalases, KatG and KatE, are the predominant scavengers of hydrogen peroxide in the environment, and the peroxiredoxin, AhpCF, plays a minor role (*Seaver and Imlay, 2001*). *E. coli* JI377, a *KatG KatE AhpCF* triple null mutant strain which cannot scavenge any hydrogen peroxide from the environment (*Seaver and Imlay, 2001*), did not protect *C. elegans* from 1 mM hydrogen peroxide killing (*Figure 8B*), whereas the *E. coli* MG1655 parental wild-type strain was protective (*Figure 8B*). We propose that *E. coli* protects *C. elegans* from hydrogen peroxide killing because it expresses catalases that efficiently deplete hydrogen peroxide from the environment, creating a local environment where hydrogen peroxide is not a threat to *C. elegans*.

To determine whether DAF-7 regulates *C. elegans* hydrogen peroxide resistance, similar to its effects on tert-butyl hydroperoxide resistance, we examined resistance to hydrogen peroxide in *daf-7* mutants. In assays with the catalase mutant *E. coli* JI377 strain, we found that *daf-7(ok3125)* increased the nematode's hydrogen peroxide resistance over two-fold relative to wild-type nematodes (*Figure 8B* and *Supplementary file 8*). ASI-ablation also increased hydrogen peroxide resistance in assays with *E. coli* JI377 (*Figure 8—figure supplement 1A* and *Supplementary file 8*). We propose that in response to TAX-4-dependent sensory perception of *E. coli*, the ASI sensory neurons express DAF-7/TGFβ to instruct target tissues to downregulate their hydrogen peroxide defenses.

Last, we investigated the possibility that reducing DAF-7-pathway signaling protects *C. elegans* from hydrogen peroxide killing via a hydrogen peroxide defense mechanism orthologous to the one by which *E. coli* protects *C. elegans*. The *C. elegans* genome contains three catalase genes in tandem—two-newly duplicated cytosolic catalases, *ctl-1* and *ctl-3*, and a peroxisomal catalase, *ctl-2*—which are the nematode orthologues of the two *E. coli* catalases, KatG and KatE (*Petriv and Rachubinski, 2004*). In our mRNA-seq analysis we found that *ctl-1* and c*tl-2* were induced by the *daf-7 (ok3125)* mutation (*Figure 8—figure supplement 1B*). In addition, we expected the *C. elegans* catalase genes to be upregulated in response to reduced DAF-7 signaling, because all three catalase genes have DAF-16 and SKN-1 binding sites in their promoters (*An and Blackwell, 2003*; *Park et al., 2009*; *Petriv and Rachubinski, 2004*), and their mRNA and protein expression increase in a DAF-16-dependent manner when DAF-2 signaling is reduced (*Dong et al., 2007*; *McElwee et al., 2003*; *Murphy et al., 2003*). To determine whether endogenous catalases could protect *C. elegans* from hydrogen peroxide when *E. coli* is not able to deplete hydrogen peroxide from the environment, we examined the effects of simultaneously increasing the dosage of all three catalase genes. We found that *ctl-1/2/3* overexpression, which increases catalase activity ten-fold (*Doonan et al., 2008*), more than doubled *C. elegans* hydrogen peroxide resistance in assays with *E. coli* JI377 (*Figure 8C* and *Supplementary file 8*). To investigate whether one of the endogenous catalases might mediate the increased hydrogen peroxide resistance of nematodes with reduced DAF-7-pathway signaling, we constructed double mutants between *daf-1* and individual catalase genes. We found that the cytosolic catalase *ctl-1(ok1242)* null mutation abrogated much of the increase in hydrogen peroxide resistance of *daf-1(m40)* mutants in assays with *E. coli* JI377 (*Figure 8D* and *Supplementary file 8*), but the peroxisomal catalase *ctl-2(ok1137)* null mutation did not (*Figure 8—figure supplement 1C* and *Supplementary file 8*). Therefore, the increase in hydrogen peroxide resistance of *daf-1* mutants is mediated in part by the CTL-1 cytosolic catalase.

In line with this functional dependence, *ctl-1* mRNA levels were elevated up to two-fold in *daf-7 (ok3125)* and *daf-1(m40)* mutant adults grown on *E. coli* OP50 (*Figure 8E*). This upregulation was partially DAF-16-dependent, since the *daf-16(mu86)* mutation caused a small but statistically

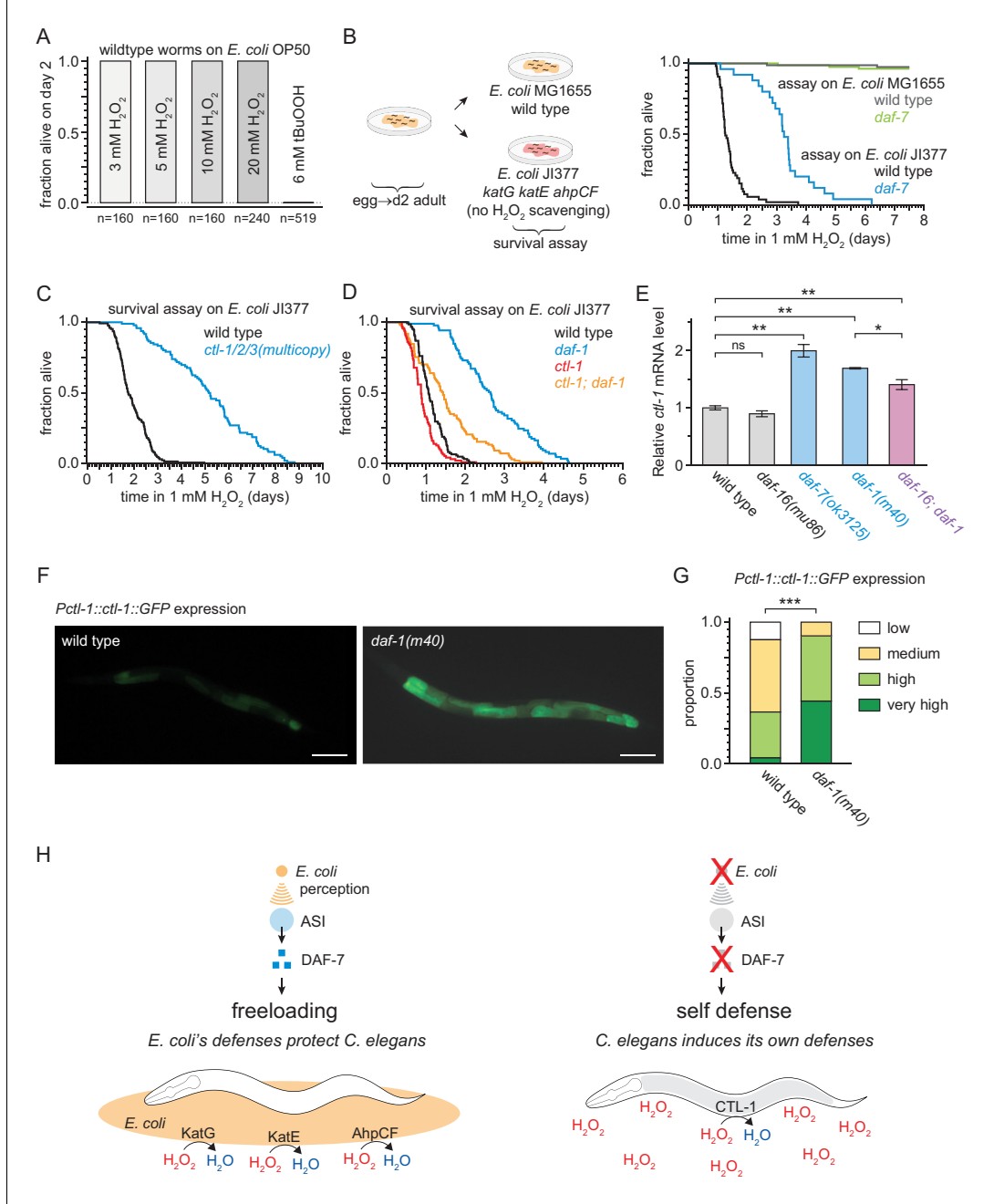

**Figure 8.** DAF-7/TGFβ signals that hydrogen-peroxide protection will be provided by catalases from *E. coli* and not by catalases from *C. elegans*. (A) *C. elegans* was sensitive to killing by tert-butyl hydroperoxide (tBuOOH), but not by hydrogen peroxide, in the presence of *E. coli* OP50. (B) Hydrogen peroxide resistance of wild type and *daf-7(ok3125) C. elegans* in assays with wild type and *Kat⁻ Ahp⁻ E. coli*. (C) Overexpression of the three endogenous catalases protects nematodes from hydrogen peroxide in assays with *Kat⁻ Ahp⁻ E. coli*. (D) The cytosolic catalase *ctl-1(ok1242)* mutation suppressed part of the increased hydrogen peroxide resistance of *daf-1(m40)* in assays with *Kat⁻ Ahp⁻ E. coli*. (E) The DAF-7/TGFβ-pathway regulates *ctl-1* mRNA expression via DAF-16/FOXO, determined by quantitative RT-PCR. Data are represented as mean ± s.e.m of three independent biological replicates, each with three technical replicates. For comparisons of *ctl-1* mRNA expression between pairs of genotypes, ** indicates $p < 0.001$, * indicates $p < 0.05$, and 'ns' indicates $p > 0.05$ (Turkey HSD test). (F) Representative pictures of the expression of the *chIs166[Pctl-1::ctl-1::gfp]* reporter in wild type animals (left picture; category: medium) or *daf-1(m40)* mutants (right picture; category: very high). Scale bar = 100 μm. (G) The expression of the promoter of *ctl-1* fused with GFP (*chIs166[Pctl-1::ctl-1::gfp]*) is higher in *daf-1(m40)* mutants (237 animals) than in wild type animals (145 animals), *** indicates $p < 0.0001$ (ordinal logistic regression). Scoring is described in Materials and methods. See *Figure 8—figure supplement 1D* for representative pictures of each expression category. (H) DAF-7/TGFβ signaling enables *C. elegans* to decide whether to induce its own hydrogen-

*Figure 8 continued on next page*

*Figure 8 continued*

peroxide defenses or, instead, freeload on protection provided by molecularly orthologous hydrogen-peroxide defenses from *E. coli*. See also
**Figure 8—figure supplement 1**. Additional statistical analyses are in **Supplementary file 8**.

The online version of this article includes the following figure supplement(s) for figure 8:

**Figure supplement 1.** DAF-7/TGFβ signals that hydrogen-peroxide protection will be provided by catalases from *E. coli* and not by catalases from *C. elegans*.

significant reduction in *ctl-1* mRNA expression in *daf-1(m40)* mutants but not in wild-type animals (**Figure 8E**). The *ctl-1* gene product is expressed only in the intestine (**Hamaguchi et al., 2019**), and this expression was elevated in *daf-1(m40)* mutants (**Figure 8F–G**, and **Figure 8—figure supplement 1D**). Taken together, these findings suggest that the DAF-7/TGFβ-pathway downregulates catalase gene expression in the intestine, partly via DAF-16. We propose that DAF-7/TGFβ signaling enables *C. elegans* to decide whether to induce its own hydrogen peroxide degrading catalases or, instead, freeload on protection provided by molecularly orthologous catalases from *E. coli* (**Figure 8H**).

## Discussion

Across the tree of life, life forms use hydrogen peroxide as an offensive weapon (**Avery and Morgan, 1924**; **Imlay, 2019**). Prevention and repair of the damage that hydrogen peroxide inflicts on macromolecules are critical for cellular health and survival (**Chance et al., 1979**). In this study, we found that in the nematode *C. elegans*, these protective responses are repressed in response to signals perceived by the nervous system. To our knowledge, the findings described here provide the first evidence of a multicellular organism modulating its defenses when it expects to freeload from the protection provided by molecularly orthologous defenses from individuals of a different species.

### Signals from sensory neurons regulate *C. elegans* hydrogen peroxide defenses

We show here that sensory neurons regulate how long *C. elegans* nematodes can survive in the presence of environmental peroxides. Peroxide resistance was higher in nematodes with a global impairment in sensory perception (**Figure 1A**). Using a systematic neuron-specific genetic-ablation approach, we identified ten classes of sensory neurons that influence the nematode's peroxide resistance, including seven classes of neurons that normally decrease peroxide resistance and three classes of neurons that normally increase it (**Figure 1B**). Why do so many neurons influence *C. elegans* peroxide resistance? One possibility is that these neurons respond to environmental cues correlated with the threat of hydrogen peroxide.

Perception of water-soluble attractants by the amphid sensory neurons ASI, ASG, and ASK—neurons that when ablated caused some of the largest increases in peroxide resistance—helps *C. elegans* navigate towards bacteria (**Bargmann and Horvitz, 1991a**), its natural food source. We found that *E. coli*, the nematode's food in laboratory experiments, influences the nematode's hydrogen peroxide resistance in three different ways. *E. coli* perception induces the expression in ASI sensory neurons of DAF-7/TGFβ (**Chang et al., 2006**; **Entchev et al., 2015**; **Gallagher et al., 2013**; **Ren et al., 1996**), a hormone that decreases the nematode's hydrogen peroxide defenses. In addition, *E. coli* ingestion increases *C. elegans* peroxide resistance in a DAF-3/CoSMAD-independent manner (**Figure 7D**). Last, *E. coli* expression of scavenging enzymes degrades hydrogen peroxide in the nematode's environment. We propose that the control of organismic peroxide resistance by neurons that sense bacteria enables nematodes to turn down their peroxide self-defenses when they sense bacteria they deem protective.

### The bacterial community influences the strategic choice between hydrogen peroxide self-defense and freeloading

We show here that *C. elegans* is safe from hydrogen peroxide attack when *E. coli* is abundant because hydrogen peroxide degrading enzymes from *E. coli* protect *C. elegans*. These *E. coli* self-defense mechanisms create a public good (**West et al., 2006**), an environment safe from the threat of hydrogen peroxide, that benefits both *E. coli* and *C. elegans*. *C. elegans* freeloads off the

hydrogen peroxide self-defense mechanisms from *E. coli* (*Figure 8H*), because it uses a public good created by *E. coli*.

*E. coli* degrades hydrogen peroxide in the environment primarily by expressing two catalases, KatG and KatE, as these enzymes account for over 95% of *E. coli's* hydrogen peroxide degrading capacity (*Seaver and Imlay, 2001*). Catalase-positive *E. coli* can protect catalase-deficient *E. coli* from hydrogen peroxide (*Ma and Eaton, 1992*). This facilitative relationship, where one species creates an environment that promotes the survival of another (*Bronstein, 2009*), also occurs across bacterial species in diverse environments: in dental plaque in the human mouth, *Actinomyces naeslundii* protects catalase-deficient *Streptococcus gordonii* by removing hydrogen peroxide (*Jakubovics et al., 2008*) and, in marine environments, catalase-positive bacteria protect the catalase-deficient cyanobacterium *Prochlorococcus*, the major photosynthetic organism in the open ocean (*Zinser, 2018*).

Unlike catalase-deficient bacteria receiving hydrogen peroxide protection from surrounding bacteria, *C. elegans* is not catalase deficient. In *C. elegans*, TAX-4-dependent sensory perception of *E. coli* stimulates the expression of DAF-7 in ASI (*Chang et al., 2006*; *Entchev et al., 2015*; *Gallagher et al., 2013*; *Ren et al., 1996*). We found that when DAF-7 signaling is reduced, target tissues induce defense mechanisms that protect *C. elegans* from hydrogen peroxide. These mechanisms are mediated in part by the DAF-16-dependent expression in the intestine of the cytosolic catalase CTL-1. Consistent with this regulatory logic, the total catalase specific activity of *C. elegans* extracts increases with decreasing concentrations of *E. coli* in the nematode's surrounding environment (*Houthoofd et al., 2002*). We propose that the TGFβ-insulin/IGF1 signaling hormonal relay that begins with DAF-7 secretion from ASI enables this sensory neuron to communicate to target tissues that they do not need to induce CTL-1 and other hydrogen peroxide protection services because *E. coli* in the surrounding environment likely provide protection by expressing orthologous hydrogen peroxide degrading enzymes. Thus, this sensory circuit enables nematodes to choose between hydrogen peroxide self-defense and freeloading strategies (*Figure 8H*).

In the complex and variable habitat where *C. elegans* lives, deciding whether to induce hydrogen peroxide defenses may be challenging. *C. elegans* cells manage this challenge by relinquishing control of their cellular hydrogen peroxide defenses to a neuronal circuit in the nematode's brain. This circuit might be able to integrate a wider variety of inputs than individual cells could, enabling a better assessment of the threat of hydrogen peroxide and precise regulation of hydrogen peroxide protective defenses.

## Coordination of behavior, development, and physiology in response to the perceived threat of hydrogen peroxide

Is the choice between hydrogen peroxide self-defense and freeloading strategies regulated by DAF-7 limited to inducing hydrogen peroxide protection services in target tissues? We favor an alternative possibility, that DAF-7 coordinates the induction of a broad phenotypic response to the perceived threat of hydrogen peroxide, because the phenotypic responses to lower DAF-7 signaling follow the expected desirable outcomes for animals that anticipate exposure to hydrogen peroxide: (i) re-routing development to form hydrogen peroxide resistant dauer larva (*Riddle and Albert, 1997*); (ii) reducing proliferation of germline stem cells (*Dalfó et al., 2012*), to prevent hydrogen peroxide induced damage to their DNA (*Wyatt et al., 2017*; *Zong et al., 2014*), (iii) reducing oocyte fertilization and egg-laying (*McKnight et al., 2014*; *Trent, 1982*), to increase the chances of progeny survival; (iv) reducing feeding (*Greer et al., 2008*), since many pathogenic bacteria produce hydrogen peroxide; (v) avoiding high oxygen concentrations (*Chang et al., 2006*), which are oxidizing; and (vi) increasing the nematode's hydrogen peroxide resistance.

These diverse phenotypic responses might be triggered by different DAF-7 levels, reflecting the adaptive benefit of reducing the harm of hydrogen peroxide in each case. Perhaps for this reason, the DAF-7 signal is relayed via different circuits to target tissues mediating some of those responses. The DAF-1 receptor and the DAF-3/DAF-5 complex function in the somatic gonad to regulate germ-cell proliferation (*Dalfó et al., 2012*), and in RIM and RIC interneurons to regulate feeding, fat storage, egg laying, and dauer-larva formation (*Greer et al., 2008*). In contrast, to regulate hydrogen peroxide resistance, DAF-1 functions in at least three different sets of interneurons (*Figure 3L*). One set includes RIM interneurons, and another comprises only the two AVK interneurons, which are not involved in regulating feeding, egg laying, and dauer-larva formation via DAF-1 signaling

(*Greer et al., 2008*). The more complex role of interneuronal DAF-1 signaling in regulating hydrogen peroxide resistance suggests that *C. elegans* takes great care to avoid inducing hydrogen peroxide protection services in target tissues unless DAF-7 levels are low.

## When do animals choose between freeloading and self-defense strategies?

Our studies provide a template for understanding how complex animals coordinate cellular hydrogen peroxide defenses. We identify sensory neurons that respond to bacterial cues as important regulators of hydrogen peroxide protection by *C. elegans* target tissues. Similar regulatory systems may exist in other animals. In mice, sensory neurons involved in pain perception respond to cues from *Staphylococcus aureus* by releasing neuropeptides that inhibit the activation of hydrogen peroxide producing immune cells (*Chiu et al., 2013*), and some of the neuropeptides secreted by these sensory neurons, including galanin and calcitonin gene-related peptide, also induce hydrogen peroxide protection in target cells (*Cui et al., 2010*; *Tullio et al., 2017*). Assigning control of cellular defenses to dedicated sensory circuits may represent a general cellular-coordination tactic used by animals to regulate induction of self-defenses for hydrogen peroxide and perhaps other threats.

We show that the two ASI amphid sensory neurons use a multistep signal relay to control the extent to which target tissues protect *C. elegans* from hydrogen peroxide. In insects and mammals, TGFβ and insulin/IGF1 signaling components regulate cellular antioxidant defenses (*Brunet et al., 2004*; *Clancy et al., 2001*; *Holzenberger et al., 2003*; *Kayanoki et al., 1994*; *Liu et al., 2012*; *Tatar et al., 2003*), so it will be interesting to determine if a conserved hormonal relay controls hydrogen peroxide defenses in all animals.

While a freeloading strategy may provide maximum fitness by inactivating self-defenses in environments where hydrogen peroxide is not a threat, this strategy need not provide maximum health or longevity to the organism. Consistent with this, in addition to lowering peroxide resistance in *C. elegans*, the ASI, ASG, and AWC amphid sensory neurons also shorten this organism's lifespan in environments with no hydrogen peroxide (*Alcedo and Kenyon, 2004*), and DAF-7/TGFβ signaling from ASI also shortens *C. elegans* lifespan in those environments (*Shaw et al., 2007*). Because sensory perception and catalases also determine health and longevity in invertebrate and vertebrate animals (*Apfeld and Kenyon, 1999*; *Libert et al., 2007*; *Pérez-Estrada et al., 2019*; *Riera et al., 2014*; *Shaw et al., 2007*), it is likely that sensory modulation presents a promising approach to induce latent defenses that could increase health and longevity in all animals.

## Materials and methods

### *C. elegans* culture, strains, and transgenes

Wild-type *C. elegans* was Bristol N2. *C. elegans* were cultured on NGM agar plates seeded with *E. coli* OP50, unless noted otherwise. For a list of all bacterial strains used in this study, see *Supplementary file 9* (*Brenner, 1974*; *Seaver and Imlay, 2001*; *Kamath et al., 2001*). For a list of all worm strains used in this study, see *Supplementary file 10* (*Chang et al., 2003*; *Chang et al., 2006*; *Wragg et al., 2007*; *Greer et al., 2008*; *Beverly et al., 2011*; *Chang et al., 2011*; *Cornils et al., 2011*; *Glauser et al., 2011*; *Dalfó et al., 2012*; *Lee et al., 2012*; *Srinivasan et al., 2012*; *Yoshida et al., 2012*; *Fernandes de Abreu et al., 2014*; *Russell et al., 2014*; *Kaplan et al., 2015*; *Vidal-Gadea et al., 2015*; *Krzyzanowski et al., 2016*; *Fletcher and Kim, 2017*; *Horspool and Chang, 2017*; *Juozaityte et al., 2017*; *Hamaguchi et al., 2019*). Double and triple mutant worms were generated by standard genetic methods. For a list of PCR genotyping primers and enzymes, and phenotypes used for strain construction, see *Supplementary file 11*. The *Ptdc-1::daf-3(+)::GFP* (pKA533) and *Pdaf-1::daf-3(+):: GFP* (pKA534) plasmids (kindly provided by Kaveh Ashrafi) were injected at 30 ng/μl into *daf-1(m40) IV; daf-3(mgDf90) X* with 20 ng/μl *Pmyo-2::RFP* and 20 ng/μl *Punc-122::DsRed*, respectively, as co-injection markers.

### Survival assays

Automated survival assays were conducted using a *C. elegans* lifespan machine scanner cluster (*Stroustrup et al., 2013*). This platform enables the acquisition of survival curves with very high

temporal resolution and large population sizes. All chemicals were obtained from Sigma. For hydrogen peroxide, tert-butyl hydroperoxide, sodium arsenite, paraquat, and dithiothreitol assays, the compound was added to molten agar immediately before pouring onto 50 mm NGM agar plates. Plates were dried (*Stroustrup et al., 2013*) and seeded with 100 µl of concentrated *E. coli* OP50 resuspended at an $OD_{600}$ of 20 (*Entchev et al., 2015*). For RNAi experiments, the appropriate *E. coli* HT115 (DE3) strain was used instead. For hydrogen peroxide assays, *E. coli* MG1655 or JI377 were used instead (*Seaver and Imlay, 2001*). Nematodes were cultured at 20°C until the onset of adulthood, and then cultured at 25°C—to potentially enhance *daf-7* mutant phenotypes (*Ren et al., 1996*; *Shaw et al., 2007*)—in groups of up to 100, on plates with 10 µg/ml 5-fluoro-2-deoxyuridine (FUDR), to avoid vulval rupture (*Leiser et al., 2016*), prevent matricidal effects of *daf-7* pathway mutants (*Shaw et al., 2007*), and eliminate live progeny. As an alternative to FUDR, we inhibited formation of the eggshell of fertilized *C. elegans* embryos with RNAi of *egg-5* (*Entchev et al., 2015*), with identical results (*Figure 2—figure supplements 1A and C-D*, and *Supplementary file 2*). For experiments with *daf-1; daf-2* double mutants, which only develop as dauers at 20°C, all strains were grown at 15°C instead of 20°C until the onset of adulthood. For food-conditioning experiments, *E. coli* OP50 was resuspended in S Basal containing streptomycin (50 µg/ml) and seeded onto plates supplemented with both streptomycin and carbenicillin, each at 50 µg/ml, as described (*Entchev et al., 2015*). For *daf-1*, *daf-3*, and *daf-16* transgenic-rescue experiments, we picked only nematodes exhibiting bright expression of the respective GFP-fusion proteins. Day two adults were transferred to lifespan machine assay plates. A typical experiment consisted of up to four genotypes or conditions, with four assay plates of each genotype or condition, each assay plate containing a maximum of 40 nematodes, and 16 assay plates housed in the same scanner. All experiments were repeated at least once, yielding the same results. Scanner temperature was calibrated to 25°C with a thermocouple (ThermoWorks USB-REF) on the bottom of an empty assay plate. Death times were automatically detected by the lifespan machine's image-analysis pipeline, with manual curation of each death time through visual inspection of all collected image data (*Stroustrup et al., 2013*), without knowledge of genotype or experimental condition.

## RNA interference

*E. coli* HT115 (DE3) bacteria with plasmids expressing double stranded RNA targeting specific genes were obtained from the Ahringer and Vidal libraries (*Kamath et al., 2001*; *Rual et al., 2004*). Empty vector plasmid pL4440 was used as control. Bacterial cultures were grown in LB broth with 100 µg/ml ampicillin at 37°C, induced with 0.1 M isopropyl-thiogalactopyranoside (IPTG) at 37°C for 4 hr, concentrated to an $OD_{600}$ of 20, and seeded onto NGM agar plates containing 50 µg/ml carbenicillin and 2 mM IPTG.

## Quantitative RT-PCR

Total RNA was extracted from day two adult animals that were transferred at the L4 stage onto NGM agar plates with 10 µg/ml FUDR seeded with *E. coli* OP50 and grown at 25°C. RNA extraction and cDNA preparation were performed as described (*Amrit et al., 2019*). Quantitative RT-PCRs were performed using the Biorad CFX Connect machine. PCR reactions were undertaken in 96-well optical reaction plates (Bio-Rad Hard Shell PCR Plates). A 20 µl PCR reaction was set up in each well using the SYBR PowerUp Green Master Mix (Applied Biosystems, USA) with 10 ng of the converted cDNA and 0.3 M primers. For each gene at least three independent biological samples were tested, each with three technical replicates. Primers used in this study include TTCCATTTCAAGCCTGCTC (*ctl-1* Fwd), ATAGTCTGGATCCGAAGAGG (*ctl-1* Rev), GGATTTGGACATGCTCCTC (*rpl-32* Fwd) (*Amrit et al., 2019*), and GATTCCCTTGCGGCTCTT (*rpl-32* Rev) (*Amrit et al., 2019*).

## Transcriptomic analysis

RNA for sequencing was extracted from day 2 adult animals that were transferred at the L4 stage onto NGM agar plates seeded with *E. coli* OP50 and grown at 25°C. We adapted a nematode lysis protocol (*Ly et al., 2015*) for bulk lysis to pool 30 individuals per sample in 240 µL of lysis buffer, which gave a sufficiently low variation among replicates (*Figure 6—figure supplement 1A*). This enabled us to collect three independent batches of three biological replicates per genotype, one batch per week, yielding a total of nine biological replicates for each genotype. cDNA preparation

from mRNA was performed by SmartSeq2 as described (*Picelli et al., 2014*). cDNA was purified using Agencourt AMPure XP magnetic beads. Nextera sequencing libraries were prepared according to the manufacturer's protocol and purified twice with Agencourt AMPure XP magnetic beads. Paired-end libraries were sequenced on an Illumina HiSeq2500 with a read length of 50 bases and approx. $2.0 \times 10^6$ reads per sample. RNA-seq reads were aligned to the *C. elegans* Wormbase reference genome (release WS265) using STAR version 2.6.0 c (*Dobin et al., 2013*) and quantified using featureCounts version 2.0.0 (*Liao et al., 2014*), both using default settings. The reads count matrix was normalized using scran (*Lun et al., 2016*). Principal component analysis was performed on the log(normalized counts + 0.5) matrix with centring and no scaling. Differential analysis was performed using a negative binomial generalized linear model as implemented by DESeq2 (*Love et al., 2014*) to compare *daf-7(ok3125)* mutants against wildtype. A batch replicate term was added to the regression equation to control for confounding. To access the expression of catalases genes (*ctl-1, ctl-2* and *ctl-3*) genome coverage of all reads was computed using BEDTools (*Quinlan and Hall, 2010*). Moreover, the multi-mapped read counts for all catalases were estimated using feature-Counts with the -M option. Resulting counts were then compared using DESseq2 as described above. Gene functional enrichments were determined by using the WormBase Enrichment Suite (*Angeles-Albores et al., 2016*). We clustered and plotted GO terms with q-value < 0.001 using REVIGO (*Supek et al., 2011*). Curated gene expression data sets were obtained from WormExp (*Yang et al., 2016*).

## Microscopy

Transgenic animals expressing a Bxy-CTL-1::GFP fusion under the control of the *C. elegans ctl-1* promoter (*Hamaguchi et al., 2019*) were scored at the young-adult stage using a fluorescence dissection stereomicroscope (Zeiss Discovery V12) under 100x magnification, following a scheme previously used to score a *gcs-1p::GFP* reporter with a similar pattern of intestinal expression (*Wang et al., 2010*). Low: only anterior or posterior intestine with patches of GFP. Medium: anterior and posterior intestine with patches GFP, middle of the intestine with dim GFP. High: anterior and posterior intestine with non-patchy GFP expression, middle of the intestine with patchy or dim GFP. Very high: strong and non-patchy GFP expression throughout the intestine. Fluorescence imaging was conducted as previously described (*Romero-Aristizabal et al., 2014*) with an Axioskop 2 FS plus microscope (Zeiss) equipped with a D470/20x excitation filter, a 500dcxr dichroic mirror, and a HQ535/50 m emission filter (all from Chroma), using a Plan-Apochromat 10 × 0.45 NA 2 mm working distance objective lens (1063–139, Zeiss). Young adult worms were placed on petri plates with modified Nematode Growth Media (to minimize background fluorescence) containing 6 mM levamisole to immobilize the animals (*Romero-Aristizabal et al., 2014*). Images were acquired with a Cool SNAP HQ$^2$ 14-bit camera (Photometrics) at 4 × 4 binning and 20 ms exposure. We performed background subtraction by removing the mode intensity value of the entire image from each pixel. This procedure removes the background due to the agar and the camera noise, since most pixels in our images were part of the background. All microscopy was performed at 22˚C.

## Behavioral assays

Pharyngeal pumping was assayed for 30 s on day 2 adults at 25˚C using a dissecting microscope under 100x magnification.

## Statistical analysis

Statistical analyses were performed in JMP Pro version 14 (SAS). Survival curves were calculated using the Kaplan-Meier method. We used the log-rank test to determine if the survival functions of two or more groups were equal. For pumping-period assays, we used the Tukey HSD post-hoc test to determine which pairs of groups in the sample differ. We used ANOVA to determine whether the fold-change in gene expression of specific gene sets and of all genes were equal. For intestinal GFP expression assays, we used ordinal logistic regression to determine if expression levels were equal between groups.

## Acknowledgements

We thank Jennifer Whangbo, Phyllis Strauss, Veronica Godoy, and Erin Cram for critical reading and detailed comments on our manuscript. Joy Alcedo, Kaveh Ashrafi, Ryan Baugh, Howard Chang, Denise Ferkey, Takaaki Hirotsu, Koichi Hasegawa, Jane Hubbard, James Imlay, Charlotte Kelley, Dennis Kim, Junho Lee, Andres Maricq, Roger Pocock, Piali Sengupta, Young-Jai You, and Yun Zhang kindly provided strains and plasmids. We benefitted from discussions with members of Erin Cram's lab, Veronica Godoy, Edward Geisinger, and Yunrong Chai. Some strains were provided by the CGC, which is funded by NIH Office of Research Infrastructure Programs (P40 OD010440), and the National BioResources Project, Japan. The research was supported by National Science Foundation CAREER grant 1750065 to JA, a Northeastern University Tier one award to JA, a National Institutes of Health grant R01AG051659 to AG, the MEIC Excelencia award BFU2017-88615-P to NS, the Spanish Ministry of Economy, Industry and Competitiveness (MEIC) to the EMBL partnership to NS, the Centro de Excelencia Severo Ochoa to NS, and the CERCA Programme/Generalitat de Catalunya, and European Research Council (ERC) under the European Union's Horizon 2020 research and innovation programme (Grant agreement No. 852201) to NS.

## Additional information

### Funding

| Funder | Grant reference number | Author |
| --- | --- | --- |
| National Science Foundation | 1750065 | Javier Apfeld |
| National Institutes of Health | R01AG051659 | Arjumand Ghazi |
| Northeastern University | Tier 1 award | Javier Apfeld |
| Spanish Ministry of Science and Innovation | BFU2017-88615-P | Nicholas Stroustrup |
| European Research Council | 852201 | Nicholas Stroustrup |
| Spanish Ministry of Science and Innovation | BES-2017-081169 | Nicholas Stroustrup |

The funders had no role in study design, data collection and interpretation, or the decision to submit the work for publication.

### Author contributions

Jodie A Schiffer, Francesco A Servello, Conceptualization, Data curation, Investigation, Visualization, Formal analysis, Methodology, Supervision, Project administration, Writing - original draft, Writing - review and editing; William R Heath, Data curation, Investigation, Methodology, Supervision, Project administration; Francis Raj Gandhi Amrit, Data curation, Investigation, Formal analysis, Methodology; Stephanie V Stumbur, Data curation, Investigation, Methodology; Matthias Eder, Data curation, Investigation, Formal analysis.; Olivier MF Martin, Data curation, Investigation, Formal analysis, Visualization; Sean B Johnsen, Julian A Stanley, Hannah Tam, Investigation, Methodology; Sarah J Brennan, Natalie G McGowan, Abigail L Vogelaar, William T Serkin, Investigation; Yuyan Xu, Validation, Investigation; Arjumand Ghazi, Data curation, Formal analysis, Supervision, Funding acquisition, Investigation, Project administration, Writing - review and editing; Nicholas Stroustrup, Data curation, Formal analysis, Supervision, Funding acquisition, Investigation, Methodology, Project administration, Writing - review and editing; Javier Apfeld, Conceptualization, Data curation, Formal analysis, Supervision, Funding acquisition, Investigation, Visualization, Methodology, Writing - original draft, Project administration, Writing - review and editing

### Author ORCIDs

Julian A Stanley http://orcid.org/0000-0002-9193-3791
Nicholas Stroustrup https://orcid.org/0000-0001-9530-7301
Javier Apfeld https://orcid.org/0000-0001-9897-5671

Decision letter and Author response
Decision letter https://doi.org/10.7554/eLife.56186.sa1
Author response https://doi.org/10.7554/eLife.56186.sa2

## Additional files

### Supplementary files

- Supplementary file 1. Statistical analysis for *Figure 1* and *Figure 1—figure supplement 1*.

- Supplementary file 2. Statistical analysis for *Figure 2* and *Figure 2—figure supplement 1*.

- Supplementary file 3. Statistical analysis for *Figure 3* and *Figure 3—figure supplement 1*.

- Supplementary file 4. Statistical analysis for *Figure 4* and *Figure 4—figure supplement 1*.

- Supplementary file 5. Statistical analysis for *Figure 5* and *Figure 5—figure supplement 1*.

- Supplementary file 6. Statistical analysis for *Figure 6* and *Figure 6—figure supplement 1*.

- Supplementary file 7. Statistical analysis for *Figure 7* and *Figure 7—figure supplement 1*.

- Supplementary file 8. Statistical analysis for *Figure 8* and *Figure 8—figure supplement 1*.

- Supplementary file 9. Bacterial strains.

- Supplementary file 10. *C. elegans* strains.

- Supplementary file 11. PCR genotyping primers and enzymes, and phenotypes used for strain construction.

- Supplementary file 12. Key resources table.

- Transparent reporting form

### Data availability

Raw mRNA-seq read files were made available under Bioproject PRJNA630551 . All data generated or analysed during this study are included in the manuscript and supporting files.

The following dataset was generated:

| Author(s) | Year | Dataset title | Dataset URL | Database and Identifier |
|---|---|---|---|---|
| Martin OMF, Eder M, Stroustrup N | 2020 | Gene expression in adult Caenorhabditis elegans daf-7 null mutants | https://www.ncbi.nlm.nih.gov/bioproject/PRJNA630551 | NCBI BioProject, PRJNA630551 |

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
