## [Decision Letter]

Thank you for submitting your article "*Caenorhabditis elegans* processes sensory information to choose between freeloading and self-defense strategies" for consideration by *eLife*. Your article has been reviewed by three peer reviewers, and the evaluation has been overseen by Oliver Hobert as Reviewing Editor and Ronald Calabrese as the Senior Editor. The reviewers have opted to remain anonymous.

As you will see below, all three reviewers were very positive about the manuscript and we would gladly consider a revised manuscript that I do not expect to have to send back to reviewers. In this revised manuscript, we ask you to please consider all the editorial suggestions made by the reviewers (to which all have agreed after discussions among themselves). The two experiments suggested by reviewer #1 and #2 are not required, but feel to discuss points related to these experiments.

Reviewer #1:

In their manuscript "*Caenorhabditis elegans* processes sensory information to choose between freeloading and self-defense Strategies", the authors discover that even when exposed to high levels of external hydrogen peroxide, the nematode *C. elegans* actively inhibits induction of its own hydrogen peroxide defense machinery if it senses the presence of bacteria.

In a series of elegant experiments, the authors demonstrate that the sensory neuron ASI plays a critical role in the regulation of peroxide resistance: In the presence of food, DAF-7/TGF-β is released from ASI and mediates inhibition of DAF-3 via DAF-1 in several interneurons. This in turn reduces DAF-3-mediated hormonal release that ultimately results in reduced DAF-16 signaling-mediated induction of peroxide defense responses (i.e. expression of catalases) in the intestine and potentially neurons.

The authors suggest that in the presence of *E. coli*, the nematode *C. elegans* relies on the bacteria's ability to reduce peroxide toxicity, and as a result actively inhibits expression of its own catalases to ultimately avoid the energetic cost of unneeded protection.

Overall, the manuscript is well written, the experiments well thought-out, thoroughly analyzed and displayed in a consistent way, which makes this manuscript an easy, enjoyable and interesting read. Moreover, the authors' findings highlight the complex interactions between organisms and their environment and demonstrate how an animal is able to adapt its survival strategies depending on its current environment.

There is only one set of experiments that I am not sure how to interpret and might need further experiments/discussions:

1) The authors demonstrate that 2 days of starvation prior to performing the peroxide survival assay significantly reduces survival. How do the authors explain this phenotype with regards to their model? These previously starved animals are exposed to the same levels of peroxide as their fed controls, because bacterial breakdown of peroxide should not be affected by the nutritional history of the worms. In addition, due to the lack of food, DAF-16-mediated induction of *C.elegans'* catalase expression should allow a faster, more robust breakdown of peroxide in starved animals compared to fed controls, who will shut down their own peroxide defense response. The authors should more clearly discuss this apparent incongruity (Are there other reasons why relatively long-term starved animals might be more sensitive?). Moreover, the authors should determine ctl-1 mRNA levels in starved vs. fed animals in absence and presence of tBuOOH.

2) The authors should think about highlighting the importance of well-balanced peroxide levels for the survival of the worm earlier in the manuscript (as is nicely discussed at the end of the manuscript). By not solely focusing on the detrimental effects of high levels of hydrogen peroxide, but also briefly mentioning the essential roles of hydrogen peroxide in signaling pathways earlier, their subsequent findings that *C. elegans* actively shuts down its own peroxide defense mechanisms might be less perplexing.

Reviewer #2:

This manuscript by Schiffer et al. reports a study of neuronal circuits used by *C. elegans* to modulate peroxide resistance. Using a series of genetic experiments, the authors demonstrated that several classes of sensory neurons upregulate, while several other classes downregulate peroxide resistance. The authors further demonstrated that prominent members of the latter class, the ASI neurons, signal via the DAF-7 ligand to regulate expression of catalases in a manner that involves DAF-1/DAF-3 and ILPs/DAF-2/DAF-16. The authors argue that this regulatory system matches catalase demand (to detox peroxides) with availability, because bacteria that make up *C. elegans* food possess substantial catalase activity too. Perhaps worms could eat lunch and have it too.

I found the topic of this paper interesting and appropriate for a broad journal like *eLife*. The authors amassed impressive amount of experimental evidence to support their claims. I have relatively minor comments regarding experiments and their interpretation, but do have some suggestions regarding overall organization of the paper.

Organization:

1) I encourage the authors to consider whether the overall order of presentation could be modified. As I first read the manuscript, I kept wondering "why peroxide? And why tBuOOH?" The last section of the paper addresses these questions, but do we need to wait five figures to get to the answer? Couldn't you open with at least some elements of Figure 7 to ground the study?

2) I think Introduction could benefit from editing. Instead of the second half of it being a brief restatement of the methodology and findings of the paper it should focus on setting up the context and motivating the study. Sections of the Discussion have highly relevant material and readers should not wait 20 pages to find out. Move them.

3) I recommend focusing Discussion on items directly related to key findings of the paper. I realize there is a lot of work in this paper and much to discuss, but it felt a little long.

Specific:

1) The "arsenite, paraquat, and DTT" experiments in Figure 2 are interesting and useful for making an argument that the circuits reported here are peroxides-specific. But are the tested concentrations relevant? Perhaps move into the main text the argument from Materials and methods that concentrations were chosen to shorten lifespan about as much as tBuOOH?

2) I found Figure 3C confusing – which neurons exactly constitute each set? It is difficult to keep in mind expression patterns of all drivers relevant for panels G, H, I. Perhaps reconsider organization of this Figure? Move panels J-M to a separate figure and bring into just-*daf-1*-focused figure relevant items from Figure 3—figure supplement 1.

3) Experiment in Figure 3F directed the authors to investigate neurons outside the expression pattern of osm-6. But is this equivalent to "indicating that *daf-1* functions in non-ciliated neurons"? Consider rewording, because *daf-1* function was not tested in subsets of ciliated neurons (nor do I think these experiments should be required).

4) Shouldn't Figure 3J, 3K and Figure 3—figure supplement 1F say "*::daf-3(+)*" as in the text of the paper?

5) I found the section starting paragraph three of subsection “Interneurons must reach a consensus to increase peroxide resistance in response to DAF-7/TGFβ from ASI” to be difficult to follow. The authors simply carried out a few cell-specific rescue experiments with *daf-3*, much the same as was done with *daf-1* in the top part of Figure 3. Why not just say this?

6) So far as I could tell, Figure 3 shows that daf-1 LOF increases resistance, an effect that could be suppressed by expressing *daf-1(+)* in all or subsets of the *daf-1* pattern or by *daf-3* LOF. Expressing *daf-3(+)* in all *daf-1* expressing cells restores *daf-1*-like phenotype in the wt-like *daf-1*;*daf-3*. However, driving *daf-3(+)* in one particular subset of *daf-1* cells does not make *daf-1*;*daf-3* double mutants be *daf-1*-like, while driving *daf-1(+)* in the same cells is sufficient to suppress the *daf-1* super-response.

Comparing *daf-1* and *daf-3* overexpression experiments in Figure 3, the authors conclude that "interneurons must reach a consensus". This is one possible interpretation. Another is that the effect of *daf-1* OE is at least somewhat *daf-3* independent. Any reason to rule this out? Or a technical explanation due to overexpression lines not being directly comparable.

7) Figure 4. Not only did *daf-12* not suppress *daf-1* super-response, but it enhanced it. Should this deserve an explanation or a comment? Same for germline.

8) Figure 4. I did not follow the logic of using *daf-7* in Figure 4B, but *daf-1* in C and D (E seems supplementary to me).

9) Figure 4. Probably useful to specify that mes^-1^ is being used for germline ablation.

10) Figure 4. Mention that tbh-1 will be dealt with later in the manuscript.

11) Figure 6F and associated text refer to "food perception". It was not clear to me what evidence was used to support this claim. All experiments in Figure 6 involve food deprivation. There are more direct ways to assess the contribution of food perception (placing bacteria on the top lid of a dish or using aztreonam to prevent bacterial consumption). I am not sure these experiments are needed here, but without them, what can be concluded regarding perception? If the assertion re food perception is based on published results, as suggested later in the manuscript, please clarify with references.

12) I may have missed it – what is the evidence that food ingestion negatively regulates DAF-3 (as in Figure 6F)?

Reviewer #3:

The manuscript by Apfeld and colleagues describes a role for DAF-7 in conferring resistance to hydrogen peroxide in the environment. The authors show that ablation of the ASI neurons, or ablation of other select sensory neurons, results in marked increased resistance to hydrogen peroxide. Notably, the authors show that activity of its receptor, DAF-1, need be active in only a subset of interneurons to suppress the increased resistance of *daf-1* mutants, whereas mutation of *daf-3* in any of these subsets of interneurons is not sufficient for the *daf-3* phenotype. The authors discuss how this may mimic a specific type of logic gate in signaling, and it also represents a cautionary tale for investigators satisfied with sufficiency in cell-specific rescue experiments. The authors carry out epistasis of pathways known to be downstream of *daf-7* signaling to rule out potential mechanisms involved in hydrogen peroxide resistance. Evidence is presented for the regulation of insulin peptides, acting on DAF-2, as being one mechanism by which *daf-7* regulates hydrogen peroxide resistance. The authors show that *E. coli* bacteria can provide resistance to hydrogen peroxide through expression of catalases, and that *C. elegans* catalase also plays a role in this resistance. The authors speculate that the sensing of *E. coli* bacteria may be linked to the regulation of endogenous catalase expression under the regulation of *daf-7*, with such expression upregulated in the absence of *E. coli* (and bacterial catalases). The assays are conducted with rigor, and the effects are large, with appropriate numbers and data analysis. The implications of the study, that neuronal signaling may regulate endogenous stress responses in a manner that is modulated by bacteria, is of potential general interest with implications for stress responses in evolutionarily diverse hosts.

1) I only have one suggestion: the large effects observed in *daf-7* mutants may, as the authors suggest, be due to upregulation of stress resistance genes, but with such a large effect, I also wonder if a behavioral phenotype might be involved; specifically, could *daf-7* mutants confer increased resistance by promoting hyperoxia avoidance, with increased molecular oxygen being more toxic in the presence of t-BOOH? This could be tested by examining if the resistance of *daf-7* mutants is affected by loss of O_2_ sensing, for example, as would be the case in a *daf-7*; gcy-35 double mutant.

---

## [Author Response]

Reviewer #1:[…]There is only one set of experiments that I am not sure how to interpret and might need further experiments/discussions:1) The authors demonstrate that 2 days of starvation prior to performing the peroxide survival assay significantly reduces survival. How do the authors explain this phenotype with regards to their model? These previously starved animals are exposed to the same levels of peroxide as their fed controls, because bacterial breakdown of peroxide should not be affected by the nutritional history of the worms. In addition, due to the lack of food, DAF-16-mediated induction of C.elegans' catalase expression should allow a faster, more robust breakdown of peroxide in starved animals compared to fed controls, who will shut down their own peroxide defense response. The authors should more clearly discuss this apparent incongruity (Are there other reasons why relatively long-term starved animals might be more sensitive?).

We thank the reviewer for this insightful comment. Food perception increases DAF-7 levels, which is predicted to lower peroxide resistance by turning off DAF-3. However, food ingestion also increases peroxide resistance in a DAF-3 independent manner. Thus, perception and ingestion of food have opposite effects on peroxide resistance. It is difficult to predict the behavior of this circuit, as we do not know the relative strength of these DAF-3-dependent and DAF-3-independent effects of food on peroxide resistance. Furthermore, food ingestion and *daf-3* cross-attenuate each other’s effects on peroxide resistance. This cross-inhibition suggests that this circuit may have switch-like properties and, therefore, the output of this circuit may depend on its initial conditions. It is also possible, as the reviewer suggests, that other mechanisms may come into play when animals are starved for an extended period.

We have rewritten and split the third paragraph of the Discussion to clarify the influence of *E. coli* on peroxide resistance. The rewritten section reads:

“Perception of water-soluble attractants by the amphid sensory neurons ASI, ASG, and ASK—neurons that when ablated caused some of the largest increases in peroxide resistance—helps *C. elegans* navigate towards bacteria (Bargmann and Horvitz, 1991a), its natural food source. […] *C. elegans* freeloads off the hydrogen peroxide self-defense mechanisms from *E. coli* (Figure 8H), because it uses a public good created by *E. coli*.”

Moreover, the authors should determine ctl-1 mRNA levels in starved vs. fed animals in absence and presence of tBuOOH.

While we have not performed these experiments, a similar experiment was reported previously by Jacques Vanfleteren’s group. They found that catalase specific activity increases in extracts of worms that are cultured at lower *E. coli* concentrations (in monoxenic liquid culture). We have added a sentence discussing those findings, which are consistent with the model we proposed. The new sentence reads:

“Consistent with this regulatory logic, the total catalase specific activity of *C. elegans* extracts increases with decreasing concentrations of *E. coli* in the nematode’s surrounding environment (Houthoofd et al., 2002).”

2) The authors should think about highlighting the importance of well-balanced peroxide levels for the survival of the worm earlier in the manuscript (as is nicely discussed at the end of the manuscript). By not solely focusing on the detrimental effects of high levels of hydrogen peroxide, but also briefly mentioning the essential roles of hydrogen peroxide in signaling pathways earlier, their subsequent findings that C. elegans actively shuts down its own peroxide defense mechanisms might be less perplexing.

As suggested by this reviewer and reviewer #2, we have moved and slightly edited two paragraphs from the Discussion into the Introduction in order to (i) describe the natural history of the threat of hydrogen peroxide to *C. elegans*, and (ii) highlight early in the paper an important challenge the animal faces: inducing protective defenses help the animal prevent and repair peroxide-induce damage, but inducing unneeded protective defenses may lead to detrimental effects. The new introductory paragraphs are:

“…In the present study, we used *C. elegans* as a model system to explore whether hydrogen peroxide protective defenses are coordinated across cells. […]

We set out to investigate whether sensory neurons coordinate hydrogen peroxide protective defenses across cells because…”

Reviewer #2:This manuscript by Schiffer et al. reports a study of neuronal circuits used by C. elegans to modulate peroxide resistance. Using a series of genetic experiments, the authors demonstrated that several classes of sensory neurons upregulate, while several other classes downregulate peroxide resistance. The authors further demonstrated that prominent members of the latter class, the ASI neurons, signal via the DAF-7 ligand to regulate expression of catalases in a manner that involves DAF-1/DAF-3 and ILPs/DAF-2/DAF-16. The authors argue that this regulatory system matches catalase demand (to detox peroxides) with availability, because bacteria that make up C. elegans food possess substantial catalase activity too. Perhaps worms could eat lunch and have it too.I found the topic of this paper interesting and appropriate for a broad journal like eLife. The authors amassed impressive amount of experimental evidence to support their claims. I have relatively minor comments regarding experiments and their interpretation, but do have some suggestions regarding overall organization of the paper.Organization:1) I encourage the authors to consider whether the overall order of presentation could be modified. As I first read the manuscript, I kept wondering "why peroxide? And why tBuOOH?" The last section of the paper addresses these questions, but do we need to wait five figures to get to the answer? Couldn't you open with at least some elements of Figure 7 to ground the study?

We agree with the reviewer’s suggestion that we better explain “why hydrogen peroxide?” early in the paper. We now moved a paragraph from the Discussion into the Introduction to describe the natural history of the threat of hydrogen peroxide to *C. elegans*.

We prefer to maintain the experimental sequence of the paper, and not shift elements of Figure 7 to the beginning of the Results section. The current narrative sequence of the paper goes from answering “which” questions, to “how” questions, to a “why” question: (1) identifying which sensory neurons regulate peroxide resistance; (2) determining how one pair of sensory neurons regulates the induction of defenses in target tissues; (3) probing how those mechanisms are influenced by the environment (*E. coli* levels); (4) determining why these mechanisms are wired in to shut down the worm’s peroxide protection in response to *E. coli* perception.

In addition, we favor the historically accurate narrative of how we came into discovering that *E. coli* defenses (catalases and peroxiredoxin) protect worms from hydrogen peroxide because that discovery, while retrospectively trivial, was far from obvious. Briefly, for about a year, we were puzzled that we could not kill *C. elegans* with hydrogen peroxide, even at high concentrations above biologically plausible, and yet we could efficiently kill worms with comparable concentrations of tert-butyl hydroperoxide. We were using tBuOOH because that was the “default” peroxide that people in the field used to kill worms. Frustratingly, we could not test if the sensory signals that regulate tBuOOH resistance would protect worms from any other peroxide, as most other peroxides are highly unstable explosives. We only solved this puzzle after we had a good understanding of the answers to the “which” and “how” questions above, leading to many insightful conversations with microbiologists in our department that pointed us to a literature on the natural history of hydrogen peroxide defenses in bacteria. Only then did we realize that the *E. coli* lawn present in our assays may degrade the hydrogen peroxide we were adding to the survival-assay petri plates. In retrospect, our finding that *E. coli* enzymes can efficiently protect *C. elegans* from hydrogen peroxide helps to explain why we could not find any paper with assays in which *C. elegans* was killed with hydrogen peroxide in the presence of *E. coli*.

2) I think Introduction could benefit from editing. Instead of the second half of it being a brief restatement of the methodology and findings of the paper it should focus on setting up the context and motivating the study. Sections of the Discussion have highly relevant material and readers should not wait 20 pages to find out. Move them.

We agree with this suggestion, and with a similar comment from reviewer #1, and have made changes to the Introduction to better set up the context motivating our studies, as discussed above in our response reviewer #1’s suggestion.

3) I recommend focusing Discussion on items directly related to key findings of the paper. I realize there is a lot of work in this paper and much to discuss, but it felt a little long.

As suggested by the reviewer, we have shortened the Discussion. We trimmed the last paragraph where we discuss the implications of our study to our understanding of the regulation of aging by signals from the nervous system. We also shifted two Discussion paragraphs into the Introduction.

Specific:1) The "arsenite, paraquat, and DTT" experiments in Figure 2 are interesting and useful for making an argument that the circuits reported here are peroxides-specific. But are the tested concentrations relevant? Perhaps move into the main text the argument from Materials and methods that concentrations were chosen to shorten lifespan about as much as tBuOOH?

We now provide in the main text, instead of in the Materials and methods section, the rationale for the concentrations of arsenite, paraquat, and DTT that we used.

2) I found Figure 3C confusing – which neurons exactly constitute each set? It is difficult to keep in mind expression patterns of all drivers relevant for panels G, H, I. Perhaps reconsider organization of this Figure? Move panels J-M to a separate figure and bring into just-daf-1-focused figure relevant items from Figure 3—figure supplement 1.

We have added a statement at the beginning of that paragraph to make explicit where the reader can find the composition of each subset of neurons where we expressed *daf-1(+)* or *daf-3(+)*:

“The cells composing each of these subsets of neurons, as well as the overlap between these subsets are diagramed in Figure 3C.”

In addition, Figure 3C now shows the promoter element used to drive gene expression in those subsets of neurons.

We do not favor splitting this figure in two, because all these experiments have the same goal: to identify which cells receive the DAF-7 signal from ASI to increase peroxide resistance.

3) Experiment in Figure 3F directed the authors to investigate neurons outside the expression pattern of osm-6. But is this equivalent to "indicating that daf-1 functions in non-ciliated neurons"? Consider rewording, because daf-1 function was not tested in subsets of ciliated neurons (nor do I think these experiments should be required).

We have reworded that statement, as suggested by the reviewer. The rewritten sentence now reads:

“Reconstituting *daf-1(+)* expression in all ciliated neurons (except BAG and FLP) using the *osm-6* promoter had a minimal effect on peroxide resistance (Figure 3F and Supplementary file 3), indicating that *daf-1* functions in non-ciliated neurons”

4) Shouldn't Figure 3J, 3K and Figure 3—figure supplement 1F say "::daf-3(+)" as in the text of the paper?

We have corrected these typos.

5) I found the section starting paragraph three of subsection “Interneurons must reach a consensus to increase peroxide resistance in response to DAF-7/TGFβ from ASI” to be difficult to follow. The authors simply carried out a few cell-specific rescue experiments with daf-3, much the same as was done with daf-1 in the top part of Figure 3. Why not just say this?

We apologize this section was difficult to follow. We have made some edits to the beginning of that section to better connect the rationale for doing the *daf-3* rescue experiments with the findings of our *daf-1* rescue experiments (see the rewritten paragraph in our response to the reviewer’s next comment).

6) So far as I could tell, Figure 3 shows that daf-1 LOF increases resistance, an effect that could be suppressed by expressing daf-1(+) in all or subsets of the daf-1 pattern or by daf-3 LOF. Expressing daf-3(+) in all daf-1 expressing cells restores daf-1-like phenotype in the wt-like daf-1;daf-3. However, driving daf-3(+) in one particular subset of daf-1 cells does not make daf-1;daf-3 double mutants be daf-1-like, while driving daf-1(+) in the same cells is sufficient to suppress the daf-1 super-response.Comparing daf-1 and daf-3 overexpression experiments in Figure 3, the authors conclude that "interneurons must reach a consensus". This is one possible interpretation. Another is that the effect of daf-1 OE is at least somewhat daf-3 independent. Any reason to rule this out? Or a technical explanation due to overexpression lines not being directly comparable.

We thank the reviewer for this insightful comment. We had not considered the possibility that DAF-3 could function only in a subset of the cells where DAF-1 functions, and that other effectors may mediate DAF-1 signaling in those cells. We have rewritten this section to discuss this possibility.

The rewritten paragraph reads:

“Where does the DAF-3/coSMAD transcription factor function to promote peroxide resistance when the DAF-1/TGFβ-receptor is inactive? We expected that DAF-3 would function in the same cells as DAF-1 to regulate peroxide resistance, because both of these canonical TGFβ signal-transduction pathway components function in *tdc-1* expressing interneurons to regulate feeding, fat storage, egg laying, and dauer-larva formation (Greer et al., 2008). […] In such a scenario, other signaling molecules would transduce DAF-1 activity in *tdc-1-*expressing neurons to regulate peroxide resistance.”

7) Figure 4. Not only did daf-12 not suppress daf-1 super-response, but it enhanced it. Should this deserve an explanation or a comment? Same for germline.

The reviewer makes a good point. We have rewritten the *daf-12* section to highlight that *daf-7; daf-12* double mutants live longer than *daf-7* mutants. We now note that *daf-12(+)* limits the peroxide resistance of *daf-7* null mutants. The rewritten paragraph section reads:

“… Loss of *daf-12* suppresses the constitutive dauer-formation phenotype of *daf-7* loss-of-function mutants during development (Thomas et al., 1993), but the *daf-12(rh61rh411)* null mutation did not suppress the increased peroxide resistance of *daf-7(ok3125)* null adults (Figure 4B and Supplementary file 4). In fact, even though the *daf-12* null mutation lowered the peroxide resistance in otherwise wild-type animals, it further increased peroxide resistance in *daf-*7 mutants. We conclude that *daf-12(+)* limits the peroxide resistance of *daf-7* mutants, and that DAF-7 lowers peroxide resistance and inhibits formation of peroxide-resistant dauer larvae via separate mechanisms.”

In the case of the germline, we now conclude with the more explicit statement:

“Therefore, DAF-1 and the germline regulate peroxide resistance via independent mechanisms.”

8) Figure 4. I did not follow the logic of using daf-7 in Figure 4B, but daf-1 in C and D (E seems supplementary to me).

Whenever possible, we have used existing *daf-7* or *daf-1* double mutant strains, generously provided by many labs that have worked on the DAF-7 TGFβ signaling pathway. We prefer to keep Figure 4E in the main text, as it rules out the possibility that the germline acts upstream of a canonical TGFβ signaling pathway to regulate peroxide resistance.

9) Figure 4. Probably useful to specify that mes-1 is being used for germline ablation.

We have re-written the relevant paragraph to clarify that we used incompletely penetrant *mes^-1^* mutations to genetically ablate the germline. The rewritten paragraph section reads:

“Germline size is reduced upon DAF-7-pathway inhibition Dalfo et al., 2012(). Mutations in the *mes^-1^* gene cause about 50% of animals to become sterile adults because they fail to form the primordial germ cells during embryogenesis, while the remaining animals develop into fertile adults (Strome et al., 1995). Germline-ablated *mes-1(ok2467)* mutants showed a 57% increase in peroxide resistance compared to their fertile *mes-1(ok2467)* mutant siblings (Figure 4D and Supplementary file 4), consistent with previous studies Steinbaugh et al., 2015(). However, *daf-1(m40)* increased peroxide resistance in both germline-ablated and fertile *mes-1* mutants (Figure 4D and Supplementary file 4). In addition, *daf-3(mgDf90)* did not affect peroxide resistance in germline-ablated *mes-1* mutants (Figure 4E and Supplementary file 4). Therefore, DAF-1 and the germline regulate peroxide resistance via independent mechanisms.”

10) Figure 4. Mention that tbh-1 will be dealt with later in the manuscript.

In the introductory paragraph leading to Figure 4A we now state that:

“In this section, and later in this manuscript, we used a genetic approach to determine whether DAF-7/TGFβ signaling acts via one or more of these mechanisms to regulate the nematode’s peroxide resistance.”

11) Figure 6F and associated text refer to "food perception". It was not clear to me what evidence was used to support this claim. All experiments in Figure 6 involve food deprivation. There are more direct ways to assess the contribution of food perception (placing bacteria on the top lid of a dish or using aztreonam to prevent bacterial consumption). I am not sure these experiments are needed here, but without them, what can be concluded regarding perception? If the assertion re food perception is based on published results, as suggested later in the manuscript, please clarify with references.

In the caption for the diagram Figure 6F we provided the relevant references to support the model that food perception regulates DAF-7; we wrote: “Sensory perception of *E. coli* induces DAF-7 expression in (Chang et al., 2006; Gallagher et al., 2013) in a concentration-dependent manner (Entchev et al., 2015; Ren et al., 1996) leading to DAF-3 repression by the DAF-7 receptor DAF-1.”

As the reviewer notes, the focus of Figure 6 is on the role of food ingestion in peroxide resistance. We realize that Figure 6F represented the first place in the manuscript where we presented the evidence supporting the regulation of DAF-7 by sensory perception of *E. coli*, the worm’s food in laboratory experiments. We believe this is warranted, because in the next section of the paper we discussed that regulation (providing the thrust for the last part of the paper). In addition, the model in Figure 6F helps to summarize the opposing roles of food perception and ingestion in regulating peroxide resistance, which we emphasize in a new section in the Discussion, to address an insightful point raised by reviewer #1.

12) I may have missed it – what is the evidence that food ingestion negatively regulates DAF-3 (as in Figure 6F)?

The evidence comes from the analysis of the peroxide resistance double mutants of *daf-3* and *eat-2* (which lowers food ingestion): “Unlike *eat-2* mutants, *daf-3* null single mutants did not decrease peroxide resistance compared with wild-type animals (Figures 3A, 3B, and 6D). However, the *eat-2(ad1116)* mutation caused a larger decrease in peroxide resistance in *daf-3* mutants than in wild-type nematodes (Figure 6D and Supplementary file 6), suggesting that *daf-3(+)* promotes peroxide resistance in *eat-2* mutants.”

Reviewer #3:The manuscript by Apfeld and colleagues describes a role for DAF-7 in conferring resistance to hydrogen peroxide in the environment. The authors show that ablation of the ASI neurons, or ablation of other select sensory neurons, results in marked increased resistance to hydrogen peroxide. Notably, the authors show that activity of its receptor, DAF-1, need be active in only a subset of interneurons to suppress the increased resistance of daf-1 mutants, whereas mutation of daf-3 in any of these subsets of interneurons is not sufficient for the daf-3 phenotype. The authors discuss how this may mimic a specific type of logic gate in signaling, and it also represents a cautionary tale for investigators satisfied with sufficiency in cell-specific rescue experiments. The authors carry out epistasis of pathways known to be downstream of daf-7 signaling to rule out potential mechanisms involved in hydrogen peroxide resistance. Evidence is presented for the regulation of insulin peptides, acting on DAF-2, as being one mechanism by which daf-7 regulates hydrogen peroxide resistance. The authors show that E. coli bacteria can provide resistance to hydrogen peroxide through expression of catalases, and that C. elegans catalase also plays a role in this resistance. The authors speculate that the sensing of E. coli bacteria may be linked to the regulation of endogenous catalase expression under the regulation of daf-7, with such expression upregulated in the absence of E. coli (and bacterial catalases). The assays are conducted with rigor, and the effects are large, with appropriate numbers and data analysis. The implications of the study, that neuronal signaling may regulate endogenous stress responses in a manner that is modulated by bacteria, is of potential general interest with implications for stress responses in evolutionarily diverse hosts.1) I only have one suggestion: the large effects observed in daf-7 mutants may, as the authors suggest, be due to upregulation of stress resistance genes, but with such a large effect, I also wonder if a behavioral phenotype might be involved; specifically, could daf-7 mutants confer increased resistance by promoting hyperoxia avoidance, with increased molecular oxygen being more toxic in the presence of t-BOOH? This could be tested by examining if the resistance of daf-7 mutants is affected by loss of O2 sensing, for example, as would be the case in a daf-7; gcy-35 double mutant.

We thank the reviewer for this interesting suggestion that behavior may be an important determinant of peroxide resistance. Consistent with that suggestion, we showed that inhibition of food ingestion in *daf-1* mutants lowers their peroxide resistance, which demonstrates that behavioral responses can lead to large effects on peroxide resistance. We plan to address more systematically, in a future paper, the extent to which different behaviors and hydrogen-peroxide resistance influence one another.